# The European 2015 drought from a climatological perspective

Monica Ionita[1,2], Lena M. Tallaksen[3], Daniel G. Kingston[4], James H. Stagge[3], Gregor Laaha[5], Henny A.J. Van Lanen[6], Patrick Scholz[1], Silvia M. Chelcea[7] and Klaus Haslinger[8]

[1] Alfred Wegener Institute Helmholtz Center for Polar and Marine Research, Bremerhaven, Germany
[2] MARUM – Center for Marine Environmental Sciences, University of Bremen, Bremen, Germany
[3] Department of Geosciences, University of Oslo, Oslo, Norway
[4] Department of Geography, University of Otago, New Zealand
[5] University of Natural Resources and Life Sciences Vienna (BOKU), Institute of Applied Statistics and Computing, Vienna, Austria
[6] Hydrology and Quantitative Water Management Group, Wageningen University, the Netherlands
[7] National Institute of Hydrology and Water Management, Bucharest, Romania
[8] Central Institute for Meteorology and Geodynamics, Vienna, Austria

*Correspondence to*: Monica Ionita (Monica.Ionita@awi.de)

## Abstract

The summer drought of 2015 affected a large portion of continental Europe and was one of the most severe droughts in the region since the summer 2003. The summer was characterized by exceptionally high temperatures in many parts of central and eastern Europe, with daily maximum temperatures 2°C warmer than the seasonal mean (1971-2000) over most of western Europe, and more than 3°C warmer in the east. It was the hottest and driest summer over the 1950-2015 study period for an area stretching from the eastern Czech Republic to Ukraine. For Europe, as a whole, it is among the six hottest and driest summers since 1950. High evapotranspiration rates combined with a lack of precipitation affected soil moisture and vegetation and led to record low river flows in several major rivers. This paper analyses the European summer drought of 2015 from a climatological perspective, including its origin, spatial and temporal development, and a comparison with the drought of 2003. The main factors controlling the occurrence and persistence of the event are discussed: temperature and precipitation anomalies, blocking episodes and sea surface temperatures (SST). The 2015 drought developed rather rapidly over the Iberian Peninsula, France, southern Benelux and central Germany in May and reached peak intensity and spatial extent by August, affecting especially the eastern part of Europe. Over the summer period, there were four heat wave episodes, all associated with persistent blocking events. Upper level atmospheric circulation over Europe was characterized by positive 500hPa geopotential height anomalies flanked by a large negative anomaly to the north and west (i.e. over the central North Atlantic Ocean extending to northern Fennoscandia) and another centre of positive geopotential height anomalies over Greenland and northern Canada. Simultaneously, summer SST was characterized by large negative anomalies in the central North Atlantic Ocean and large positive anomalies in the Mediterranean basin. Composite analysis shows that the western Mediterranean SST is strongly related to the occurrence of dry and hot summers over the last 66 years (especially over the eastern part of Europe), whereas central North Atlantic SST is not. In accompanying papers, the hydrological perspective and impacts of the summer 2015 drought are presented. Together, these three papers summarize a collaborative initiative of members of UNESCO's FRIEND-Water program to provide a timely pan-European assessment of the 2015 summer drought.

# 1 Introduction

Drought is part of the natural climate cycle and commonly affects large areas and can last for several months or even years. It is a complex phenomenon that has wide ranging environmental and socioeconomic impacts and is globally considered to be one of the costliest natural hazards (Wilhite, 2000, EEA, 2010). In Europe, the overall losses due to drought over the period 1976 – 2006 have been estimated to about 100 billion Euros (EC, 2007). Drought affects all components of the hydrological cycle, from its origin as a deficit in precipitation, which combined with high evapotranspiration losses can lead to deficit in soil moisture and subsequently can manifest itself as a hydrological drought, i.e. deficits in streamflow and groundwater (Tallaksen and Van Lanen, 2004). In cold regions, a temperature anomaly may cause similar winter drought to develop (Van Loon and Van Lanen, 2012). Consequently, drought can cause a wide range of impacts affecting the environment, society and economy, where impacts on agriculture and public water supply are most frequently reported (Stahl et al., 2016). Prolonged droughts with severe impacts, such as the major drought in 2003, have highlighted Europe's vulnerability to this natural hazard and alerted governments, stake holders and operational agencies to the disastrous effects droughts may have on the society and economy, including the need for mitigation measures (EEA, 2001, 2010; EC, 2012). The year 2015 was characterized by one of the worst droughts recorded in Europe, particularly in the central and eastern part of the continent (Van Lanen et al., 2016). The 2015 summer (June – July – August (JJA)) was the third warmest summer in Europe since 1910 (after 2003 and 2010), and coincided with the warmest month (August 2015) and year (2015) on record, with a global temperature anomaly of 0.90°C above the 20th century average (NOAA, 2016). Record high temperatures were recorded at several places in Europe in July and August 2015, including Kitzigen (Germany) – 40.3°C; Catania (Italy) – 42.8°C; Cordoba (Spain) – 45.2°C; Dobrochovice (Chechnya) – 39.8°C and Yerevan (Armenia) – 40.9°C. In some countries (e.g. Czech Republic), it was the second driest summer of the last 50 years, where only 2003 had lower rainfall (Van Lanen et al., 2016). The drought was accompanied by extreme low flows, especially over the central and eastern part of Europe, and caused serious socio-economic impacts in various water-related sectors (Van Lanen et al., 2016).

The 2015 event was extreme, but not singular. In fact, the beginning of the 21st century has seen an increase in the frequency of prolonged droughts and heat waves in Europe, including the event of 2003 that affected large parts of central Europe. Other major events include the drought in 2008 over the Iberian Peninsula (Andreu et al., 2009), the drought and heat wave in Russia in 2010 (Barriopedro et al., 2011) and the spring drought of 2011 over western part of Europe. As average global temperatures continue to rise, it is estimated that droughts will further intensify in the 21st

century, particularly in certain seasons and regions, e.g. in southern and eastern Europe (Stagge et al., 2014), parts of North America, Central America, and southern Africa (Seneviratne et al., 2012; Orlowski and Seneviratne, 2012; Prudhomme et al., 2013; Wanders et al., 2015).

A better understanding of the spatial and temporal development of these major drought events, in particular their triggering mechanisms and persistence, is vital to enable a better prediction of prolonged dry periods and extensive drought extent (Tallaksen and Stahl, 2014; Van Huijgevoort et al., 2013). This includes the drivers of extreme drought periods and the association between droughts and climatic, oceanic and local factors such as land-atmosphere feedbacks, which may amplify the drought development (e.g. Whan et al., 2015). Persistent dry (wet) conditions are usually associated with anticyclonic (cyclonic) circulation, while the sea surface temperatures (SST) can play also an important role via the interaction with large scale climatic/oceanic modes of variability (e.g. North Atlantic oscillation (NAO) and/or El Niño-Southern Oscillation (ENSO) (Ionita et al., 2011; 2015; Schubert et al., 2014). Altogether, when favorable phase conditions are met, both large scale atmospheric as well as oceanic factors could act as precursors to dry (wet) summers over Europe (Kingston et al., 2013, 2015; Ionita et al., 2012, 2015).

This paper summarizes a collaborative initiative of members of UNESCO's FRIEND-Water programme to perform a pan-European assessment of the drought of 2015 from a climatological point of view. In an accompanying paper (Laaha et al., 2016), a similar pan-European assessment of the drought from a hydrological perspective is performed with focus on streamflow records. Impacts and examples of how the drought of 2015 was managed are described by Van Lanen et al. (2016). The objectives of this paper are: a) to characterize the temporal and spatial extent of the summer 2015 drought event using both daily and monthly climate data (surface level analyses); b) to analyze the key drivers of the event, with a special emphasis on the role played by the SST and large scale (atmospheric) circulation modes of variability (upper level analyses); c) to compare the key characteristics of the 2015 drought with the 2003 event and d) to place the summer 2015 drought event into a long-term perspective.

The paper is organized as follows: Section 2 gives an introduction to the data and methods used. The results of surface analysis of the summer drought of 2015 are shown in Section 3, while the prevailing ocean-atmosphere conditions are presented in Section 4. The similarities and differences with the summer event of 2003 and the long-term context of the 2015 event are discussed in Section 5. Concluding remarks are given in Section 6.

## 2 Data and methods

### 2.1 Climate, oceanic and hydrologic data

Climate variables, i.e. daily time series of precipitation (P), mean temperature (T) and maximum temperature (Tmax), originate from the E-OBS dataset (Haylock et al., 2008). E-OBS is a daily gridded observational dataset for precipitation, minimum, mean and maximum temperature and sea level pressure in Europe. It is based on the European Climate Assessment and dataset (ECA&D) station information. The full version dataset covers the period 1950-01-01 until 2015-12-31, in total 66 years. For this study, we use the regular 0.25° x 0.25° latitude - longitude grid.

Daily Outgoing Longwave Radiation (OLR) was extracted from NOAA Interpolated Outgoing Longwave Radiation dataset (Liebmann and Smith, 1996). For the Northern Hemisphere atmospheric circulation, we use the daily and monthly means of geopotential height at 500-hPa (Z500) and 850-hPa (Z850) levels, the zonal wind at 850-hPa (U850) level, 500-hPa (U500) level and 250 (U250) level, the meridional wind at 850-hPa (V850) level and 500-hPa (V500) level and the mean sea level pressure (SLP) from the NCEP/NCAR 40-year reanalysis project (Kalnay et al., 1996) on a 2.5° x 2.5° grid. The vertically integrated water vapor transport (WVT) (Peixoto and Oort, 1992) is calculated using zonal wind (u), meridional wind (v) and specific humidity (q), also from the NCEP/NCAR reanalysis (Kalnay et al., 1996). WVT vectors for latitude ($\phi$) and longitude ($\lambda$) are defined as follows:

$$\vec{Q}(\lambda,\phi,t) = Q_\lambda \vec{i} + Q_\phi \vec{j} \qquad \text{eq. (1)}$$

Where zonal ($Q_\lambda$) and meridional ($Q_\Phi$) components of Q are given by eq. (2):

$$Q_\lambda = \int_0^{p_0} qu \frac{dp}{g}$$
$$Q_\phi = \int_0^{p_0} qv \frac{dp}{g} \qquad \text{eq. (2)}$$

For each vertical layer and each grid-point, we calculate the product between the monthly values of horizontal wind (u) and specific humidity (q). The result is multiplied with the pressure thickness of the layer they represent and divided by gravity. The WVT is obtained by summation of water transport for all layers located between the eEarth's surface and 300 hPa level. We also used the divergence of water vapor K, which is in balance with the surface fresh water flux E-P (Peixoto and Ort, 1992):

$$\nabla \bullet \vec{Q} = E - P,$$

where $\nabla$ denotes the two dimensional divergence operator, E evaporation and P precipitation. Regions of mean positive divergence (E-P>0) constitute source regions of water vapour, whereas

regions of convergence (E-P<0) are sink regions for water vapour.

Global SST is extracted from the Extended Reconstructed Sea Surface Temperature data et (ERSSTv4b; Liu et al., 2014). This dataset covers the period 1854 − 2015 and has a spatial resolution of 2° x 2°. For daily SST, we use the OISST dataset (Reynolds at al., 2007).

## 2.2 Climatological drought indices (SPI and SPEI)

The SPI (McKee et al., 1993; Guttman, 1999) and SPEI (Vicente-Serrano et al., 2010; Beguería et al., 2013) are used as indicators for meteorological/climatological drought. The SPI has become a favored meteorological drought index in Europe and is recommended as a drought index by the

10 World Meteorological Organization (Hayes et al., 2011; WMO, 2012). The more recently developed SPEI uses a similar methodology, but includes a more comprehensive formulation of the climatic water balance which may better quantify drought (Beguería et al., 2013). The SPI and SPEI normalize accumulated precipitation (P) and climatic water balance (P − PE), respectively, where PE represents the potential evapotranspiration. By taking into account both short and long

accumulated anomalies, both indices have a multiscalar feature, and hence give the user the possibility to approximate agricultural, hydrological, and socioeconomic drought by adjusting the accumulation period of the indices for a particular region (Vicente-Serrano et al., 2011; Hayes et al., 2011). Here, we use an accumulation period of three months (91 days) for both indices, hereafter referred to as SPI3 and SPEI3, both derived relative to the reference period 1971-2000. The index

values represent the number of standard deviations from typical conditions for a given location and time of year and therefore allow for objective, relative comparisons across locations with different climatologies and highly non-normal precipitation distributions (Stagge et al., 2015). A negative SPI (SPEI) represents values lower than the median and values less than -2 are considered to represent extremely dry conditions, corresponding to a 1 in 50-year event (WMO, 2012). A 3-

25 months accumulation period was chosen to capture the seasonal development of the drought and enable comparison with monthly anomalies in P and T. Kingston et al., (2015) concluded that atmospheric anomalies associated with major drought events were very similar whether a 3 or 6-month accumulation period was used.

Potential evapotranspiration for the computation of the SPEI was calculated using the Hargreaves

equation (Hargreaves and Samani, 1985). This method uses the daily difference between the maximum temperature ($T_{max}$) and the minimum temperature ($T_{min}$) as a proxy to estimate the net radiation. Use of the Hargreaves equation when calculating SPEI has shown to be robust for Europe, providing similar results to more complex models (Stagge et al., 2014).

### 2.3 Teleconnection indices

Teleconnection indices for the North Atlantic Oscillation (NAO), the Arctic Oscillation (AO), the Scandinavian (SCA) and the East Atlantic (EA) patterns, have been downloaded from http://www.cpc.ncep.noaa.gov/data/teledoc/teleintro.shtml for the period 1950–2015. The Niño3.4 index was used to characterise ENSO, and is defined as the average of SST anomalies over the domain 170° - 120°W; 5°S - 5°N.

# 3 Climate variables and drought indices

### 3.1 Precipitation and Maximum Temperature

The summer 2015 drought and heat wave as measured by the rank maps of the monthly lowest precipitation totals and the maximum temperature (for the months May to August) relative to historical values from 1950-2015, are shown in Figures 1 and 2. The most affected regions during the summer 2015 event were the central and eastern part of Europe and the northern Balkans. In May, abnormally dry and warm conditions started to develop over the Iberian Peninsula, south-eastern France, southern Benelux and Germany (Figures 1a and S1a). May 2015 is ranked as the driest and hottest May in the 1950-2015 study period over the central and eastern Iberian Peninsula, and the second driest May over a small band covering the central part of Germany and eastern Romania (Figure 1a).

Over the Iberian Peninsula, there was a precipitation deficit of up to 70 mm/month (~20% of total precipitation) compared to the 1971-2000 average (Figure S1c). June 2015 ranks as either the driest or second driest month in the study period across the eastern part of Europe (Figure 1b). As a consequence, in June, the spatial distribution of below-normal precipitation moved north and eastwards, covering central part of Europe, the northern Balkans and parts of Belarus, Ukraine (Figure S1c). By July 2015, drought conditions were well established. The rainfall deficit over the central, southern and eastern part of Europe was more than 75 mm/month (~40% of total precipitation) (Figure S1e). July 2015 saw drought intensity slightly decreasing in many regions (Figure 1c), with a region of moderate drought in central France, but extremes in western Romania continuing from June. This eastern European centre of the drought expanded significantly in August 2015, ranking as the driest month in these regions (Figure 1d). The most affected regions with respect to a deficit in precipitation were central and eastern part of Europe, and a smaller area over Finland and the Scandinavian Peninsula (Figure S1g).

In contrast to the warming in south-western Europe in May, negative temperature anomalies were recorded in north-western Europe, with negative temperature anomalies in Scandinavia (Figure S1b). In June, major parts of the European continent experienced maximum temperatures that were

significantly above average, with anomalies of more than 3°C in an area stretching from western Spain to central and eastern France, the western Alps and Ukraine (Figure S1d). In July, the heat wave intensified with anomalies as high as 5°C over France. The area affected also expanded and became more clearly defined, stretching from Spain and France up to eastern and central part of Europe, western Ukraine and the Balkans (Figure S1f). July 2015 was the hottest July on record over Spain, large parts of Italy and Czech Republic and the second hottest July over the central and eastern part of France and in some areas in the eastern part of Europe (Figure 2c). In contrast, July 2015 was cooler than normal in north-western Europe, with up to -5°C over Fennoscandia and north-western part of Russia (Figure S1f). In August, the heat wave was no longer present over the Iberian Peninsula and southern France, but continued to develop in central and Eastern Europe, with anomalies of up to 5 °C over Poland and the western part of Ukraine (Figure S1h). August 2015 is ranked as the hottest August on record across most of Poland, Ukraine and Belarus, and the second hottest August over a broader area extending from France in the west to Belarus in the east and covering latitudes from the northern Romania to Latvia (Figure 2d). Compared to June and July, the northern part of Europe also experienced a relatively warm August (Figure S1h).

Throughout the summer, four heat wave episodes (defined as Tmax> 25 °C) can be identified in Europe when averaging the daily maximum temperature over the affected region (approximately 0 - 30°E and 40 - 55°N) (Figure S3). The highest maximum temperatures were recorded in the first 10 days of August.

### 3.2 Drought indices: the SPI and SPEI

The SPI3 and SPEI3 for June, July and August (Figure 3) show similar development in time as for the climate variables, with the area most affected shifting from south - western parts of Europe in June and July, towards the central and eastern parts of Europe in August. Although monthly precipitation over France shows an increase in August, the climatic water balance drought conditions still persist, especially over the western part of the country. SPI and SPEI values as low as -3 are recorded in August (Figure S2f), and the most extreme values were found in southern Spain, parts of France and Germany, Belarus and Western Ukraine. Similar to the P and T pattern, June - August 2015 SPI3 and SPEI3 ranks as the driest 3-months on record over regions covering the eastern part of Europe.

Figures 1 and 3 (and S1 and S3) show that the rainfall deficit and climatological drought conditions persisted for almost three months over large regions in Europe. In contrast, there were regions in Europe that received above average precipitation, especially in the northern and north-western part of Europe and Fennoscandia. In summary, the summer 2015 was characterized by a dipole-like structure with rainfall deficit (P) and climatological drought (SPI and SPEI) in the central and

southern part of Europe and positive anomalies (P and SPI and SPEI) over Fennoscandia and the British Isles. Drought conditions during the summer months of 2015 appear more spatially extended for the SPEI3 index (which incorporates also temperature information) as compared to the SPI3 index (which is solely based on precipitation). For example, July SPI3 does not show July 2015 as the driest month on record over Spain (Figure 3c), whereas the July SPEI3 does (Figure 3d). This is likely due to an overall low (3-month) seasonal precipitation in this region, which makes distinguishing between precipitation-based drought anomalies (SPI3) difficult, whereas accounting for potential evapotranspiration in the SPEI3 highlights the severity of this drought, which was ranked as the hottest July on record in Spain (Figure 2c). This implies that the SPEI is a more comprehensive measure of drought severity and is therefore used instead of the SPI to quantify meteorological drought for the remainder of this paper. Similar results have been found by Hoy et al. (2016), who showed that SPEI and the Water Balance Anomaly Index (WBAI) are a better estimate of drought severity compared to precipitation based drought indices (SPI).

## 4 Ocean-atmosphere conditions

Previous studies have emphasized that extreme temperature and precipitation deficits do not occur without similarly extreme anomalous conditions in the large scale atmospheric circulation and ocean (e.g. Black et al., 2004; Feudale and Schukla, 2007, 2010). In the following paragraphs, we present the evolution of the monthly SST anomalies over the Atlantic Ocean and Mediterranean Sea, the Z500 anomalies and the OLR anomalies between May 2015 and August 2015, and discuss how these may be linked to the anomalies observed in climate variables and drought indices.

### 4.1 Sea surface temperature

Figure 4 shows the evolution of the SST anomalies between May 2015 and August 2015. The corresponding rank maps are shown in Figure S4. At the beginning of May, the Mediterranean Sea was up to 1°C warmer compared to climatology. At the same time, the central North Atlantic Ocean (south of Greenland) was colder by around 0.5°C. The North Atlantic Ocean was characterized by three distinct anomalous SST centers: i) warm SSTs in the central Atlantic Ocean extending from the east coast of the U.S. up to the western coast of southern Europe and the Mediterranean Sea; ii) cold SSTs south-east of Greenland and iii) warm SSTs poleward of 65°N (Figure 4a). In terms of extremes, May 2015 was the 6[th] coldest May over a small region in the central North Atlantic Ocean (Figure S4a) and the second warmest May over the western part of the Mediterranean Sea (Figure S4b). In June, the SST pattern persisted and the anomalous SST anomalies south-east of Greenland and in the Mediterranean Sea intensified (Figure 4b). The cold blob in the North Atlantic Ocean is ranked as the coldest June over the last 154 years and the western part of the Mediterranean Sea is ranked as the fourth warmest June (Figure S4c and S4d, respectively). In July,

the warm anomaly in the Mediterranean Sea is further intensified (Figure 4c), with SST values exceeding 28°C (Figure S5). The extreme warm anomaly in July is consistent with the strong extreme maximum temperature anomalies recoded over most of Europe in July (Figure 2c). The warm anomaly in the Mediterranean Sea and the Atlantic Ocean in the 20°N - 40°N band was accompanied by a similar strong negative anomaly of ~1.5°C in the northern part of the Atlantic Ocean (Figure 4c). The western part of the Mediterranean Sea ranks as the warmest July on record. Record breaking SSTs also occurred in the eastern central Atlantic (Figure S4f). In August, the warmth in the Mediterranean Sea and the Atlantic Basin in the 20°N - 40°N band and the cooling south of Greenland persisted, with almost the same amplitude (Figure 4d). The altering (positive – negative – positive) spatial pattern in the SST anomalies was present throughout the summer in the Atlantic Ocean basin, suggesting that the air-sea interaction associated with the northward shift of the subtropical high plays an important role (Czaja and Frankignoul, 2002; Huang and Shukla, 2005). The SSTs in the Mediterranean Sea were again in the top eight warmest August on record (Figure S4g). The SSTs in the Atlantic basin, centered around the 30°N band, rank in the first three warmest August (Figure S4g). In contrast to this, the small region in the central North Atlantic Basin ranks in the first eight coldest August (Figure S4f).

In the following, we investigate more closely the evolution of the daily SST anomalies that might have influenced the European climate throughout the summer 2015, either directly or via teleconnections. First, we have computed two-time series of daily SST anomalies, to encapsulate the two key anomaly areas: the spatial average over the central North Atlantic Ocean (-30°E - -10°E; 50°N - 60°N) and over the western Mediterranean Sea (0°E - 25°E; 30°N - 45°N). These two regions are chosen based on the rank maps in Figure S4. In a similar manner, we computed a daily SPEI3 index, averaged over the area affected the most by the drought conditions in summer 2015 (18°E - 32°E; 48°N - 52°N). This area was chosen based on the ranking map for SPEI3 (June-August; Figure 3f) and comprises the region which ranks as the driest on record.

Figure 5a shows the daily SST evolution from January to December 2015. A pronounced and abrupt increase in the SST anomalies over the Mediterranean Sea took place at the end of June 2015 over a period of ~3 weeks, when there was a marked increase in the SST of ~2°C accompanied by a similar abrupt cooling in the central North Atlantic Ocean of ~1°C. As demonstrated in Figure 5a, the SST anomalies in the Mediterranean Sea are accompanied by similar SST anomalies in the central North Atlantic Ocean, but of opposite sign throughout 2015. A similar dipole-like structure of warm Mediterranean Sea – cold central North Atlantic Ocean was found to be involved in triggering the heat waves in summer 2003 (Feudale and Shukla, 2007, 2010). In terms of long term anomalies, summer of 2015 stands out as 3[rd] coldest summer over the central North Atlantic Ocean (Figure 10c) and the 3[rd] warmest one in the western Mediterranean Sea (Figure 10d). The warmest

SSTs in the Mediterranean Sea (2003, 2010 and 2015) over the last 160 years occurred in connection with extreme heat waves and large-scale droughts in Europe. The daily evolution of the western Mediterranean Sea SST index and the daily SPEI index is shown in Figure 5b. By applying a cross-correlation analysis between the two-time series, it has been observed that the highest correlation (r = -0.8, 99% significance level) is found when the western Mediterranean Sea SST index leads the SPEI3 index by ~25 days. Similar analysis has been done for the SPEI3 daily index and the North Atlantic Ocean SST index, but the correlation is much smaller (r = 0.4, when the North Atlantic Ocean SST index leads the SPEI3 index by ~50 days). A more detailed discussion regarding the influence of these two particular regions (western Mediterranean Sea SST vs. North Atlantic Ocean SST) on the occurrence of dry summer over the eastern part of Europe is given in Section 5.3.

## 4.2 Geopotential height at 500hPa and the outgoing longwave radiation (OLR)

During May 2015, Fennoscandia and the British Isles were under the influence of cyclonic conditions, whereas the Iberian Peninsula and the south-western part of France were affected by anticyclonic conditions (Figures 6a and S7a) and increased OLR anomalies associated with reduced cloud cover and precipitation over these regions (Figure 6b). In June 2015, the anticyclonic circulation extended from the eastern North Atlantic Ocean up to central Europe, Belarus, Ukraine and the Balkans, while the cyclonic flow extended from the North Atlantic Ocean over Fennoscandia up to the northern part of Russia (Figures 6c and S2b). These Z500 anomalies were accompanied by increased OLR anomalies over France, Ukraine and the Mediterranean Sea (Figure 6d). In June and July 2015, there was an obvious wave-train of altering Z500 anomalies: negative, but weak, Z500 anomalies over central-northern Canada, positive Z500 anomalies over Greenland, negative Z500 anomalies over Northern Atlantic Basin and Fennoscandia and positive Z500 anomalies over central and southern part of Europe (Figure 6c and 6e). This wave-train was accompanied by excessive precipitation (e.g. Figure 3b) and reduced OLR anomalies over Scandinavia and the British Isles, and heat waves, dryness and increased OLR anomalies over the Iberian Peninsula, central Europe and the Balkans.

In August 2015, the Z500 anomalies are projecting onto an $\Omega$-like block pattern with positive Z500 anomalies over Alaska and Greenland, followed by negative Z500 anomalies in the middle of the North Atlantic Ocean and an anomalous positive center over Europe (Figure 6g). The anomalous Z500 center over Europe suggests a dominant subsidence and adiabatic motion associated with reduced cloudiness and increased incoming solar radiation (Figure 6h).

Since there were four consecutive heat wave episodes (Figure S3) with small interruptions in between, daily evolution of the atmospheric circulation and WVT averaged over the heat wave

episodes were examined (Figure 7). All four heat wave episodes were associated with atmospheric blocking situations dominated by a south-westerly flow regime and a northward shift of the storm-tracks (red arrows in Figure 7 – left panels). The highest maximum temperatures occurred, in each of the four cases, near the center of the block, where descending motions and reduced cloudiness contributed to anomalously warm temperatures and dryness. To the east of the heat wave region, anomalously cold temperatures occurred in conjunction with an upper level trough and cold air intrusions from the north. During the first two heat wave episodes (29.06 – 08.07, 15.07 – 26.07), the vector plots of the WVT show a clear deflection of the storm-track away from the central and southern part of Europe (Figure 7b and 7d). The axis of the WVT is directed through the British Isles and Fennoscandia, in agreement with the positive precipitation anomalies and low temperatures recorded over these areas in June and July (Figure S1). The two episodes recorded in August (01.08 – 16.08 and 26.08 – 31.08) show a distinct pattern with the axis of the WVT deflected northwards, resulting in an extended area (whole Europe and Fennoscandia) affected by positive temperature anomalies and precipitation deficit (Figure 7f and 7h). In agreement with these findings, Ionita et al. (2012) have shown that periods characterized by low streamflow anomalies over the Rhine River catchment area are associated with a north-ward shift of the Atlantic storm-tracks, whereas periods with high streamflow anomalies are associated with a more zonal circulation and with moisture inflow from the Atlantic Basin.

The large-scale Z500 anomalies identified throughout the 2015 summer had a similar structure to the low level anomalies (Figure S6), indicating an approximately equivalent barotropic vertical structure. Moreover, the wave - train like pattern of alternating Z500 anomalies and the streamfunction anomalies at 850hPa (Figure S7) suggest a Rossby wave signal propagating from U.S. to Russia. These findings are consistent with the findings of Schubert et al. (2014), which have shown that droughts and heat waves over the Eurasian continent often are associated with the occurrence of stationary Rossby waves. For example, the 2003 European and 2010 Russian heat waves were driven mainly by a stationary Rossby wave that extended across the northern Eurasia (Schubert et al., 2011). Under the influence of $\Omega$ blocks and northward shifted storm tracks, warm dry air from southern Europe and Africa was pulled northward, pushing temperatures higher than normal over the Iberian Peninsula, central Europe and the Balkans.

The heat waves and associated blocking episodes and the northward shift of the storm-tracks throughout the 2015 summer, might have been caused by unusual warm SSTs in the Mediterranean Sea (Figures 4 and 5). A similar situation was observed in the summer 2003. Modeling results (Feudale and Shukla, 2007, 2010) indicate that exceptionally warm SSTs over the Mediterranean Sea and the central North Atlantic Ocean could be responsible for shifting the jet stream northwards, leaving Europe under drought conditions and heat waves.

**4.3 Teleconnection patterns**

Previous studies have shown the importance of teleconnection patterns in modulating the climate of Europe in different seasons (Hurrell, 1995; Trigo, 2004; Andrade et al., 2012; Casanueva et al., 2014; Ionita et al., 2014; 2015). Here, we investigate the role of these teleconnection patterns in modulating/influencing the drought conditions in summer 2015. The first eight months of 2015 were characterized by altering phases of some teleconnection patterns or by a persistent positive or negative phase. In 2015, a very strong El Niño event developed (Figure 8a). Since there is no clear evidence regarding the influence of El Niño/ La Niña events on the European climate, it is rather difficult to estimate the impact of this strong event on the occurrence of the extreme drought and heat waves in summer 2015. Overall, the strongest influence of ENSO has been found to be during the fall season (Shaman, 2014). Moreover, the influence of ENSO on the precipitation over Europe has been found to vary strongly on a seasonal basis (Mariotti et al., 2002), but for the summer season no significant relationship has previously been observed.

The teleconnection patterns shown to have the strongest impact on European precipitation and temperature, and hence drought conditions, are the NAO, the EA and the SCA teleconnection patterns (Hurrell, 1995; Barnston and Livezey, 1987; Comas-Bru and McDermott, 2014; Ionita et al., 2015; Kingston et al., 2015). The NAO was in a positive phase from January until June 2015, followed by an abrupt shift towards the negative phase in July and August (Figure 8b). In July 2015, the NAO recorded the lowest value of -3.14 (for the month of July) over the last 65 years (Figure S8b). The negative NAO phase during the summer months was accompanied by a positive phase of the EA pattern (Figure 8c) and a negative phase of the SCA pattern (Figure 8d). The recorded values of the SCA index were among the lowest over the last 65 years for the months of May and June 2015 (May: -2.15 and June: -1.52, Figure S8d). In summer, a negative NAO is associated with reduced precipitation and high temperatures over the Mediterranean region and the Balkans (Blade et al., 2012). Moreover, the summer NAO has been found to strongly affect the Mediterranean Sea SSTs. A negative summer NAO is associated with anomalously warm SSTs in the Mediterranean basin (Blade et al., 2012), in agreement with our findings from Section 3.3. As such, the negative phase of NAO and SCA during the summer months could partially explain the reduced precipitation and high temperatures over the Iberian Peninsula, the central part of Europe, and the Balkans, while simultaneously producing increased precipitation amounts and low temperatures over the Northern part of Europe in 2015 summer. This spatial pattern is also seen in a recent study by Casanueva et al. (2014), who find that the main drivers of the summer precipitation over Europe are NAO and SCA and that the northern part of Europe receives more precipitation during the negative phase of SCA and NAO, whereas the rest of Europe receives less precipitation and is more exposed to droughts.

## 5 Long-term context of the summer 2015 drought event

### 5.1 Similarities/discrepancies with summer 2003

Since there are indications of an increase in the frequency of future heat waves and droughts in multiple European regions (Christensen et al., 2007; Orlowski and Seneviratne, 2012; Prudhomme et al., 2013; Van Huijgevoort et al., 2014; Giuntoli et al., 2015; Wanders et al., 2015), it is important to analyze the most extreme events at a pan-European scale and to study the underlying processes in a consistent manner. One way to tackle this topic is to analyze the precursors and background mechanisms between different extreme events to search for similarities and/or discrepancies. The summer drought of 2015 is one of the worst droughts since the event of 2003. Summer 2003 was notable for three reasons: i) the surface temperature anomalies associated with the heat wave were more than five standard deviations above the mean in some parts of Europe (Schär et al., 2004); ii) the high precipitation deficit and evapotranspiration losses led to water shortages, temporal cessation of agricultural activities and even the temporary shutdown of various power plants (Stahl et al., 2016) and iii) it caused around 70.000 heat-related deaths, mainly in the western and central part of Europe (Robine et al., 2008).

Spring and summer of 2003 were characterized by reduced precipitation over most of the central and southern part of Europe (Figures S9a and S10a). The precipitation deficit became visible at the end of winter season. Summer 2003 was characterized by a very dispersed pattern, with relatively small regions affected by record-breaking precipitation deficits (Figure 9a). Contrary to 2003, winter 2015 was rather wet over the areas affected later by droughts in summer and the precipitation deficit only became detectable at the end of spring (over the Iberian Peninsula, France and Germany; Figure S9b). Summer 2015 was further characterized by a more homogenous deficit pattern, with extreme precipitation deficits over large areas covering the eastern part of Europe (Figure 9b).

The year 2003 experienced the warmest summer on record over most of central Europe, including north-western part of Spain, France, Italy, Germany, Switzerland, Austria, and the western part of Czech Republic (Figure 9c). Positive anomalies in maximum temperature (~2°C), were seen first in spring over the Iberian Peninsula, France, the British Isles, Benelux and Germany (Figure S9c). This pattern amplified in summer, peaking in August, when Europe registered the warmest summer in the last 500 years (Luterbacher et al., 2004), with record-breaking temperature in the western and central part of Europe (Figure S10c). In spring 2015, positive maximum temperature anomalies were recorded over the Iberian Peninsula (~2°C), France and the northern part of Fennoscandia (Figure S9d). However, the affected area was smaller than in spring 2003. In summer 2015, the situation changed considerably, with the central and southern part of Europe being affected by heat waves and maximum temperature anomalies of up to ~4°C (Figure S10d). Summer 2015 was the

warmest on record over small regions in the south and western part of Spain, the western part of Ukraine and the eastern part of Poland and the second warmest summer on record over the most of the central and eastern part of Europe (Figure 9d). In general, the spatial extent of the precipitation deficit and high temperatures was rather different between 2003 and 2015. In summer 2003, the most affected areas by climatological drought (anomaly in P, SPI and SPEI) and heat waves (anomaly in Tmax), were the central and western part of Europe and to some degree Scandinavia, whereas in summer 2015 the affected regions were extending from central Europe up to the east (Ukraine, Belarus).

The SPEI3 and SPI3 seasonal evolution (Figures 10 and S11), show a similar pattern as the precipitation anomalies (Figures S9 and S10). Spring 2003 was very dry over the central and southern part of Europe and this pattern persisted and accentuated in summer 2003 (Figures 10a and 10b and Figures S11a and S11b). In spring 2015, a much smaller area was affected, including the Iberian Peninsula, France and the central part of Germany (Figures S11c and S11d), whereas the northern part of Europe, Balkan and Belarus were very wet. In 2015, the drought conditions became more evident and accentuated in summer, especially over the eastern part of France, southern part of Germany, Austria and the whole eastern part of Europe (Figures 10c and 10d). The differences between the summer 2003 and 2015 drought is computed for both SPI3 and SPEI3 for the months May (Figures S11e and S11f) representing the spring season, and August (Figure 10e and 10f), reflecting the summer season. As seen from Figures S11e and S11f, spring 2003 was overall much drier compared to spring 2015 over most of the European continent. In summary, the drought of 2003 was a slow developing drought, which began much earlier in the year over most of the affected regions. It resulted in extreme high temperatures, affecting mostly western and central Europe, extending from the very south to the north. By contrast, the drought of 2015 had a much more rapid development from its initiation in spring, and a more central and eastern European location, extending from west to east.

Despite the relatively large differences in the initial meteorological conditions of the two events, the SST anomalies in summer 2003 had a similar structure to those recorded in summer 2015 (Figures 11a and 11b). The central North Atlantic Ocean was cooler than normal in both summers, although in 2003 the cold region was smaller compared to summer 2015. The cold spot in the central North Atlantic Ocean in 2003 was flanked entirely by positive SST anomalies. The main difference between summer 2003 and 2015, in terms of SST anomalies, can be observed over the North Sea. In summer 2003, the North Sea was much warmer compared to 2015. In contrast, over the Mediterranean Sea, the SSTs in both summers were among the three warmest on record (Section 3.3 and Figure 14d). According to modeling studies, a warm Mediterranean Sea alone could not produce the heat wave, but it can reinforce it (Feudale and Shukla, 2007). The local Mediterranean

SST increase may contribute to an increase in the number of hot days over the European continent through increased oceanic heat flux to the atmosphere. Moreover, for the summer 2003, the North Sea and the surrounding part of the North Atlantic Ocean were also very warm and could have played a role in the development of the 2003 heat wave by reducing the baroclinicity in the European region and enhancing blocking situations (Feudale and Shukla, 2010).

In summer 2003, the upper level atmospheric circulation was characterized by a large positive Z500 anomaly over Europe flanked by negative Z500 anomalies over the central North Atlantic Ocean and over Belarus, Ukraine and western Russia (Figure 11c). This positive Z500 anomaly over central Europe is connected to an anticyclonic circulation, indicating dominant adiabatic descending motions, which results in enhanced incoming solar radiation which in turns warms the surface. In summer 2003, the overall structure of the Z500 anomalies projects onto a typical $\Omega$ blocking structure. The spatial structure of the upper level atmospheric circulation in summer 2015 is different compared to summer 2003 (Figure 11d). In summer 2015, the upper level atmospheric circulation over the European continent was characterized by a large positive Z500 anomaly flanked by a large negative Z500 anomaly to the north and west (i.e. over the central North Atlantic Ocean extending to northern Scandinavia). The Z500 anomalies in summer 2015 resemble the Atlantic Low regime (Cassou et al., 2005). This weather regime is associated with the occurrence of extreme warm days over the western and southern part of Europe in summer, due to advection of warm air masses form northern Africa and the Mediterranean Sea.

Although the overall spatial structure of the Z500 anomalies for the two extreme summers are not similar, they are both associated with anticyclonic circulations over Europe, flanked by cyclonic circulation over the surrounding areas and with very warm Mediterranean Sea and tropical Atlantic Ocean and cold central North Atlantic Ocean. As previously shown, the anticyclonic circulation in summer over Europe could be strongly influenced by the tropical Atlantic Ocean, due to a northward shift in the Intertropical Convergence Zone (ITCZ) position (Cassou et al., 2005). Based on modelling and observational studies, it has been shown that the probability of heat waves to occur over Europe is much higher when the tropical forcing is taken into account (Cassou et al., 2005). Whereas, the summer 2003 drought and heat wave was the consequence of a prolonged dry period characterized by persistent anticyclonic circulation, which started already at the end of winter, the drought in summer 2015 originated much later, i.e. at the end of spring.

## 5.2 Comparison with other extreme dry and hot summers

To assess the severity of the summer 2015 drought and heat wave in a long-term context, we have analyzed the daily maximum temperature for JJA and SPEI3 for August over study period (1950-2015). In Figure 12 we show the six warmest (the highest daily maximum temperatures recorded at

each grid point, Figure 12a) and driest (Figure 12b) years on record and their spatial extent. It is interesting to note that five out of the six hottest years were recorded in the last 15 years. The only exception is the summer 1972. Each of these years were recorded over well-defined regions: 1972 – eastern parts of Fennoscandia and north-western parts of Russia, 2002 – central parts of Norway and Belarus, 2003 – eastern parts of the Iberian Peninsula and central parts of Europe, 2010 – the Balkan region, 2010 - western parts of Russia and 2015 – southern parts of Czech Republic and Ukraine. The summers 1972, 2003 and 2015 are affecting the largest regions in terms of extreme temperatures. In a recent publication, Russo et al. (2016) have also shown, by using a more complex definition of the heat wave concept, that the heat waves recorded in July 1972 in Finland as well as the summer 2003 heat wave, were the strongest heat waves observed in the record.

Three out of the six driest summers in terms of high climatic water deficit (as measured by the SPEI3), i.e. 1955, 1972, 1976, 2003, 2010 and 2015, occurred in the last 15 years (Figure 12b). The summers of 1972, 2003, 2010 and 2015 stand out as both very hot and dry, indicating that the high climatic water deficit (P-PE) had a strong influence on the occurrence of droughts. In contrast, the summers of 1955 and 1976 stand out as very dry, but not extremely hot, indicating that these dry summers may be driven mainly by precipitation deficits.

### 5.3 Summer droughts and large-scale atmospheric circulation in a long-term context

To examine the drivers of historical droughts, including the extreme summer 2015 event, we have created an SPEI3 August index, defined as the mean SPEI3 within the most affected region (black box in Figure 13a, Section 3.2). Using this index, we have computed composite maps showing the mean atmospheric circulation and North Atlantic SSTs for the driest summer years (regional SPEI3 index lower than 0.75 standard deviation), comprising the following years: 1951, 1952, 1953, 1959, 1963, 1964, 1976, 1983, 1992, 1994, 1995, 2002 and 2015 (Figure 13b). It is interesting to note that 2003, generally considered a benchmark western European drought, is not considered an extreme year in terms of SPEI3 for this region because it has slightly different climate drivers (see Section 5).

Dry years in this eastern European region, which was the most affected region during summer 2015, are usually associated with a tripole-like pattern of a large positive Z500 anomaly over the central and eastern part of Europe, flanked by a large negative Z500 anomaly to the north and south-east (i.e. over Greenland and Mediterranean Sea) (Figure 13c). Summer (SPEI3) droughts in this region are also associated with a northward deflection of the Atlantic storm-tracks (Figure 13d), which produces exceptionally warm summer temperatures for most of Europe, centered over the eastern Europe (Figure 13e). North Atlantic SSTs during these events are characterized by a tripole-like pattern, with altering signs of SST anomalies over the North Atlantic basin and the Mediterranean

Sea. Dry summers over the eastern part of Europe are associated with positive SST anomalies on the eastern coast of the U.S. and over the Mediterranean and North Sea, negative SST anomalies south of Greenland and positive SST anomalies over the Barents Sea (Figure 13f). It is worthwhile to note that the anomalies identified in the Z500, WVT, Tmax and SST fields (Figure 13), associated with dry summers over the eastern part of Europe, closely resemble the conditions during the 2015 event identified in the previous sections. This suggests that the 2015 drought followed a relatively typical, but extreme, climatological pattern that produces drought in eastern Europe.

Previous studies have emphasized the role of the Atlantic and Mediterranean SST in driving the occurrence of meteorological droughts and heat waves over the European regions (Feudale and Shukla, 2011a; 2011b; Ionita et al., 2011; Kingston et al., 2014). In a recent study, Duchez et al. (2016) have argued that cold SST anomalies in the central North Atlantic Ocean were the main driver of the summer 2015 heat wave over Europe, by initiating a Rossby wave train that favored the development of positive Z500 anomalies and extreme temperature over Central Europe. To test this hypothesis, we compared the same eastern Europe regional SPEI3 index against mean SST anomalies for both the central North Atlantic Ocean (black box in Figure 14a), where the coldest temperatures were recorded in summer 2015, and the western Mediterranean Sea (black box in Figure 14b), where the warmest temperatures were recorded in summer 2015. During the common period (1950-2015), there was a relatively small correlation (r = 0.12) between the North Atlantic SST index (black line) and the August SPEI3 index (red line). Both indices show the lowest values on record in 2015, but appear to be otherwise unrelated, with dry years (negative SPEI3 index) occurring equally when the central North Atlantic Ocean is cold or warm. Unlike the North Atlantic SST, the western Mediterranean SST index has a significant (r = -0.36, 99% significance level) correlation with summer SPEI3 in eastern Europe. This suggests a link between drought in eastern Europe and warm Mediterranean SST anomalies, though the strong trend in Mediterranean SST data and complex nature of atmosphere-land-ocean feedback warrants more detailed study.

To further clarify this finding, a reverse analysis was performed, generating composite maps of August SPEI3 (with one season lag and in phase) for anomalously cold North Atlantic SST and anomalously warm Mediterranean SST, both defined as ≥ 0.75 standard deviation (Figure 15). Similar composite maps for summer Tmax are provided as supplementary figures (Figure S13a). This analysis confirms that cool North Atlantic SSTs have historically had little effect on meteorological drought in the eastern European region, the most affected during the 2015 drought event (Figures 15a and 15c). The only detectable effect of cold North Atlantic SST, particularly in the spring (MAM), is to produce moderate drought over the western part of Germany and Netherlands (Figure 15a). Alternatively, warm Mediterranean SSTs have historically preceded (Figure 15b) and occurred concurrently with summer drought over much of central and eastern

Europe. The regions where the significant anomalies are observed in Figure 15d are similar to those regions affected by drought in summer 2015, further supporting the conclusion that the 2015 drought event followed a historically common (but severe) climate pattern, characterized by large positive Z500 anomaly over the central and eastern Europe and high Mediterranean SSTs. While the cool north Atlantic SSTs were historically extreme and concurrent with the 2015 drought event, there is no indication that this association has occurred in the recent historical record (60 years), creating some doubt as to its causal effect.

## 6 Discussion and conclusions

This study has presented the key drivers and characteristics of the summer 2015 drought in Europe. A series of hydroclimatological variables have been investigated to characterize the extraordinary drought of summer 2015 and its dynamic relationship with multiple causal factors, namely the large-scale atmospheric circulation and teleconnections, and SST. Summer 2015 has been identified as the fourth one in a series of extreme summers in Europe characterized by droughts and heat waves that started in 2003 (Schär et al., 2004) and continued in 2010 (Barriopedro et al., 2011) and 2013 (Dong et al., 2014). These summers were all characterized by long-lasting droughts, heat waves and record temperatures at different locations, depending on the particular event.

The largest precipitation anomalies, in summer 2015, coincided with a very persistent upper-level ridge. The regions most affected by the drought and heat waves were situated south of the axis of the North Atlantic jet stream and were under the influence of large-scale descending motion, reduced precipitation and clear skies. A particular feature of the summer 2015 was a series of four heat wave episodes, all associated with persistent blocking events. These anomalies (precipitation and temperature) are indicative of a mechanisms controlled by the radiative forcing, rather than thermal advection. Similar results have been obtained by Andrade et al. (2012), who showed that warm days in summer, over the central and southern part of Europe, are controlled by radiative forcing enhanced due to persistent high-pressure system.

Over western, central and eastern part of Europe, the high pressure system allowed hot air from the tropics to move north and get persist over these regions. Clear skies allowed for the temperatures to rise even further, creating a stronger center of high pressure, reinforcing the already stagnant $\Omega$ block atmospheric pattern. The high pressure system acted as a barrier, preventing low pressure systems from moving over Europe and pushing them instead to the north. Contrary to the situation observed over the western, central and eastern part of Europe, due to the influence of the low-pressure system in June and July, the British Isles and Fennoscandia were subjected to unstable weather conditions, low temperatures and cloud formation.

The summer SSTs averaged over the Mediterranean Sea showed that summer 2015 was the third warmest summer over the last 160 years. Although the very warm Mediterranean Sea and the occurrence of blocking patterns over Europe are likely to be closely related, the exact mechanism by which the Mediterranean SSTs could influence the atmospheric circulation over Europe is not fully understood (Beniston and Diaz, 2004). Feudale and Shukla (2007) suggested that global SSTs are responsible for the anticyclonic circulation over Europe. By prescribing just the Mediterranean SST anomaly in summer, they show that the upper level atmospheric circulation over Europe can be realistically simulated, although with smaller amplitudes than if observed SSTs are used. Opposite to these results, Jung et al. (2006) showed that enhanced SSTs over the Mediterranean Sea had only marginal influence on the mid-tropospheric atmospheric circulation over Europe. Nevertheless, it must be recognized that three of the most extreme summer SPEI3 drought and heat wave events in the last 15 years occurred simultaneously with the highest Mediterranean SSTs on record. Although the summer 2015 was also characterized by one of the coldest summers in the central North Atlantic, no significant relationship between the occurrence of dry summers over the eastern part of Europe and the central North Atlantic SSTs could be found.

In terms of hemispheric-scale teleconnections, the summer 2015 drought was characterized by a negative phase of the NAO and SCA patterns, including the lowest July value for the NAO in the last 60 years. A negative phase of the NAO and SCA is consistent with dry and warm summers over the central and eastern part of Europe and wet and relatively cold conditions over the British Isles and Fennoscandia. Specifically, the negative phase of summer NAO is associated with a southward shift in the North Atlantic storm track, which in turn brings wet and cold summers over the U.K. and Fennoscandia and dry and warm summers over the central, southern and eastern part of Europe. When comparing the drivers of summer 2003 drought/heat wave and summer 2015, some important features stand out:

> The summer 2003 drought and heat wave were caused by a northerly displacement of the Atlantic subtropical high, whereas in summer 2015 all four distinct heat wave episodes were associated with blocking situations and a wavy jet stream (Figure 7).

> The extreme temperatures in summer 2003 were amplified by a severe soil moisture deficit (Schär et al., 2004), as a consequence of a very dry and cold winter and a very dry and warm spring. In contrast to 2003, the drought in 2015 started to develop late spring. The winter and spring 2015 were normal in terms of precipitation and temperature anomalies, with small exceptions over the Iberian Peninsula. The 2015 drought developed rather rapidly over the Iberian Peninsula, southern Benelux and France in May and reached peak intensity and spatial extent by August, affecting especially the eastern part of Europe.

> The surface temperature anomalies in summer 2003 were more than 5 standard deviations above the mean in parts of Europe (Schär et al., 2004). Due to the exceptional heat wave in summer 2003, the number of fatalities was very high (~70.000). In 2015, the number of fatalities was much smaller compared to summer 2003 (~1250) (MunichRe, 2016).

> For both extreme summers, the central Atlantic Ocean was colder than normal, while the Mediterranean Sea was much warmer than normal (~3°C). Moreover, record-breaking temperatures were registered in both summers, but in different regions.

> The use of an index-based analysis and composite analysis of central North Atlantic SST and the western Mediterranean SST, on one side, and the August SPEI3 field, on the other side, are not enough to establish causality. Nevertheless, this kind of analysis has shown that warm springs/summers over the western Mediterranean Sea are usually associated with summer droughts over the eastern part of Europe. The same is not true for the central North Atlantic SST. The index based analysis and composites have shown that a cold central North Atlantic can be associated with both dry or wet summer over the eastern part of Europe.

Both extreme summers were associated with a more meandering polar jet, a circulation structure that has appeared more often during the last decades in connection with the Arctic amplification phenomena (Cohen et al., 2014). In a recent study, Lehmann and Coumou (2015) showed that changes in the mid-latitude circulation strongly affect the number and intensity of extreme events. They showed that summer heat extremes are associated with low storm track activity, due to a reduced eddy kinetic energy (EKE). Low summertime EKE is significantly linked to positive geopotential height anomalies, and hence has a strong impact on the occurrence of heat waves and droughts. An observed decreasing trend in the EKE in summer has been associated with favorable conditions for heat extremes such as the ones in 2003 and 2010 (Lehmann and Coumou, 2015). The decreasing trend in EKE is in agreement with the observed increasing trend in the frequency and persistence of summertime anticyclonic circulation patterns since 1979 over the Northern Hemisphere, especially over the U.S., Europe and western Asia (Horton et al., 2015).

In a longer-term context, summer 2015 ranks as the hottest and driest summer since 1950 over extended regions in eastern Europe. Moreover, summer 2015 ranks also among the six hottest and driest summers since 1950 at the continental scale. An interesting finding is also the fact that five (three) out of the six hottest (driest) summers since 1950 occurred after the year 2000. The summers 1972, 2003, 2010 and 2015 rank both as hot and dry, while the summers 2002 and 2012 rank just as hot, and summers 1955 and 1976 rank just as dry (out of the six hottest and driest years, respectively).

The increase in the frequency and persistence of summertime anticyclonic circulation contributed significantly to heat waves and droughts over those regions. Nevertheless, caution should be taken when applying the conclusion from this study about the possible drivers of the drought event, since there are other hemispheric/regional mechanisms that also could have played a role (e.g. a weakening of the summer atmospheric circulation (Coumou et al., 2015) and/or the Arctic Amplification (Cohen et al., 2015)). Simulations with general circulation models may be also necessary in the future to investigate the full spectrum of causes of severe drought events.

The study has further assessed the severity of the event in terms of spatial coverage and the strength of the anomaly, through the use of standardized drought indices (SPI and SPEI), and compared it with the extreme drought of 2003. One important feature of these events, namely the heat waves that accompanied the droughts, was given special attention. However, to assess the full range of drought impacts, many of which are related to a lack of water (Van Lanen et al., 2016), requires additional analyses and data such as soil moisture, groundwater and streamflow observations

Improved management of drought requires a common action of the hydrological and climatic communities that should include monitoring of hydro-meteorological variables, multi-monthly and seasonal forecasting of both climatic and hydrological variables, impact assessments and exploration of potential promising measures to reduce impacts, accounting for the specific conditions at the river basin scale. As such, drought impact and management studies require a concerted multi-disciplinary action from the climatic and hydrological communities that consider both climate and hydrological controls on drought. This paper, along with its counterparts addressing the hydrological perspective and the impacts of the European drought of 2015 (Laaha et al., 2016; Van Lanen et al., 2016), can be seen as a first effort of the climatological and hydrological communities to jointly evaluate the causes and consequence of an extreme event from both perspectives.

*Acknowledgements.* This study is promoted by Helmholtz funding through the Polar Regions and Coasts in the Changing Earth System (PACES) program of the AWI. Funding by the Helmholtz Climate Initiative REKLIM is gratefully acknowledged. The work forms a contribution to the UNESCO-IHP FRIEND-Water Programme. We acknowledge the E-OBS dataset from the EU-FP6 project ENSEMBLES (http://ensembles-eu.metoffice.com) and the data providers in the ECA&D project (http://www.ecad.eu).

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

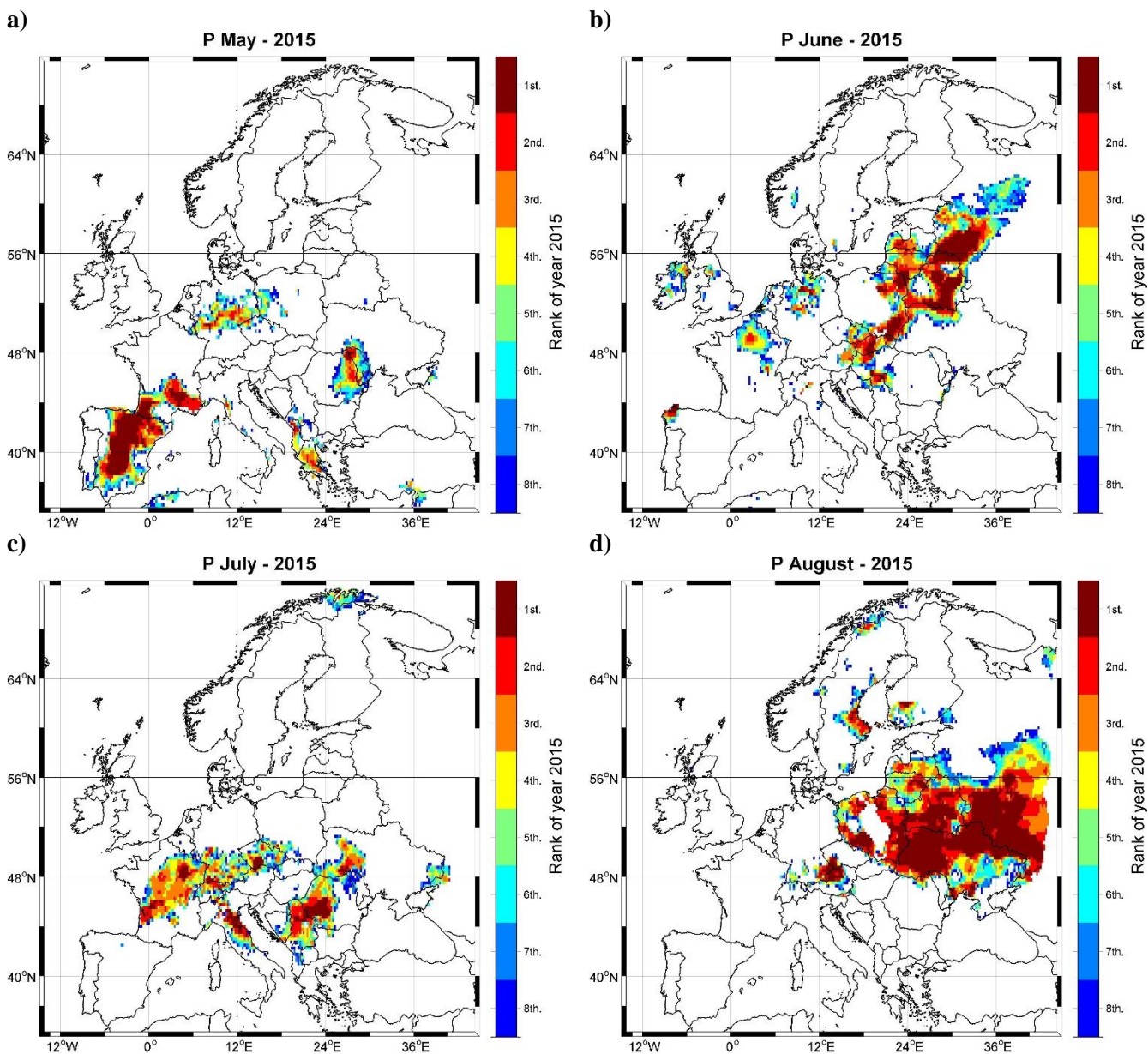

**Figure 1.** Top 8 ranking of 2015 monthly lowest P: a) May P; b) June P; c) July P and d) August P. In this figure, 1 means the driest (P) month since 1950, 2 signifies the second driest, and all ranks greater than 8 are shown in white. Analyzed period: 1950 – 2015.

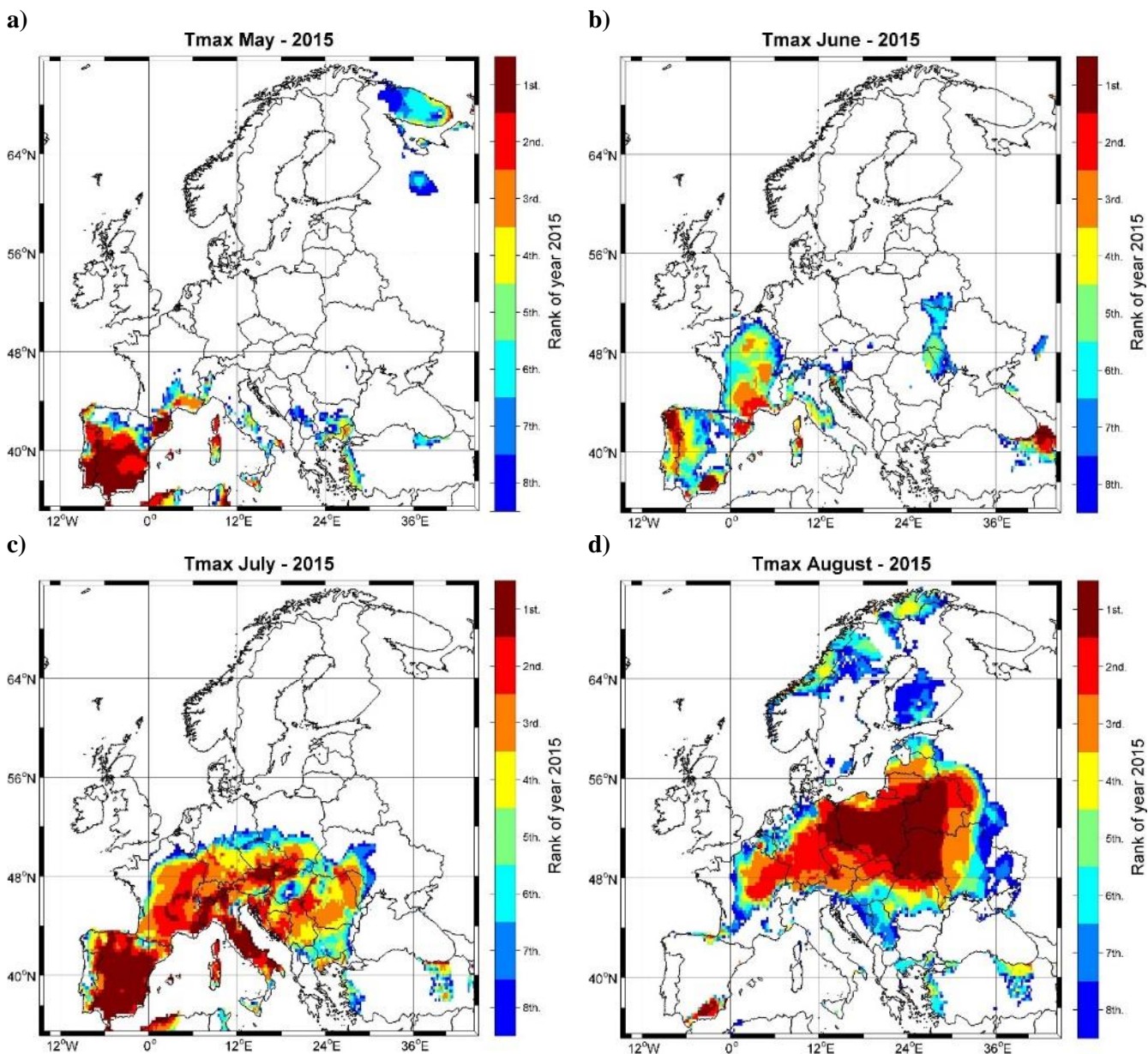

**Figure 2.** Top 8 ranking of 2015 monthly Tmax: a) May Tmax; b) June Tmax; c) July Tmax and d) August Tmax. In this figure, 1 means the warmest (TT) month since 1950, 2 signifies the second warmest, and all ranks greater than 8 are shown in white. Analyzed period: 1950 – 2015.

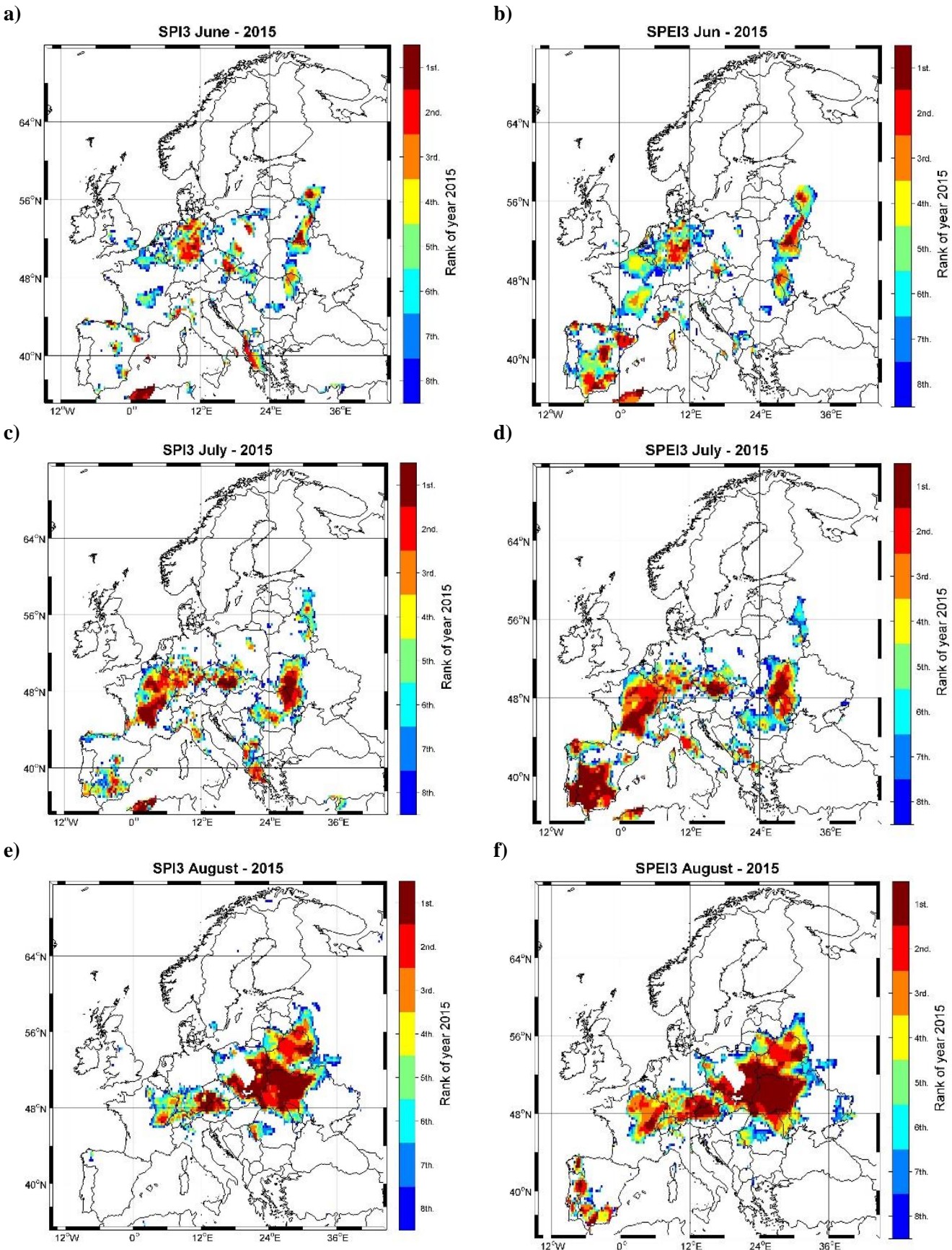

**Figure 3.** Top 8 ranking of 2015 monthly SPI3 (left column) and 2015 SPEI3 (right column): a) June SPI3; b) June SPEI3; c) July SPI3; d) July SPEI3; e) August SPI3 and f) August SPEI3. In this figure, 1 means the driest month since 1950, 2 signifies the second driest, and all ranks greater than 8 are shown in white. Analyzed period: 1950 – 2015.

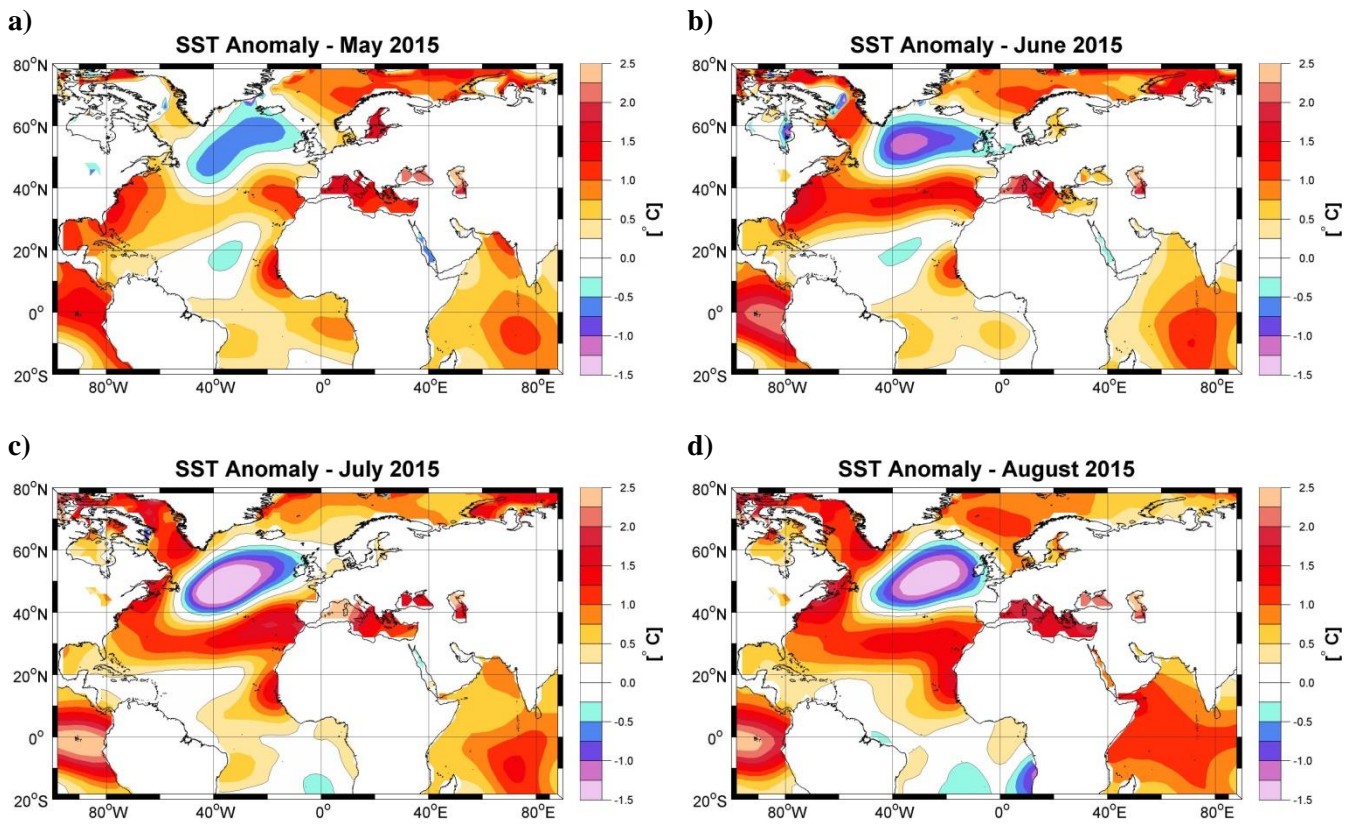

**Figure 4.** Monthly SST anomalies: a) May 2015; b) June 2015; c) July 2015 and d) August 2015. The anomalies are computed relative to the period 1971 – 2000.

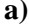

**a)**

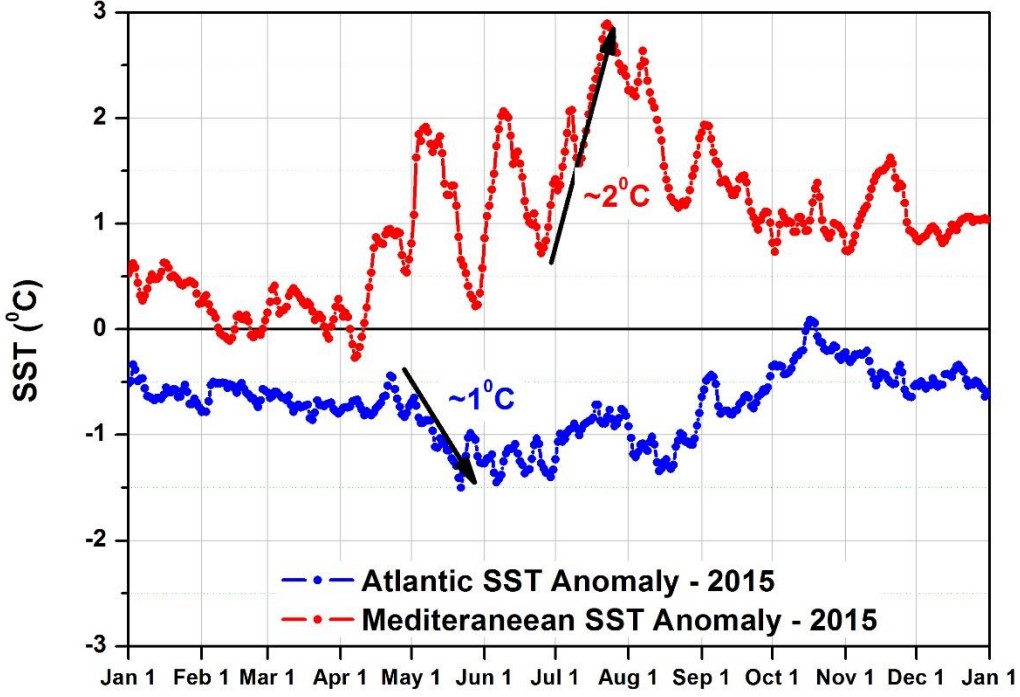

**b)**

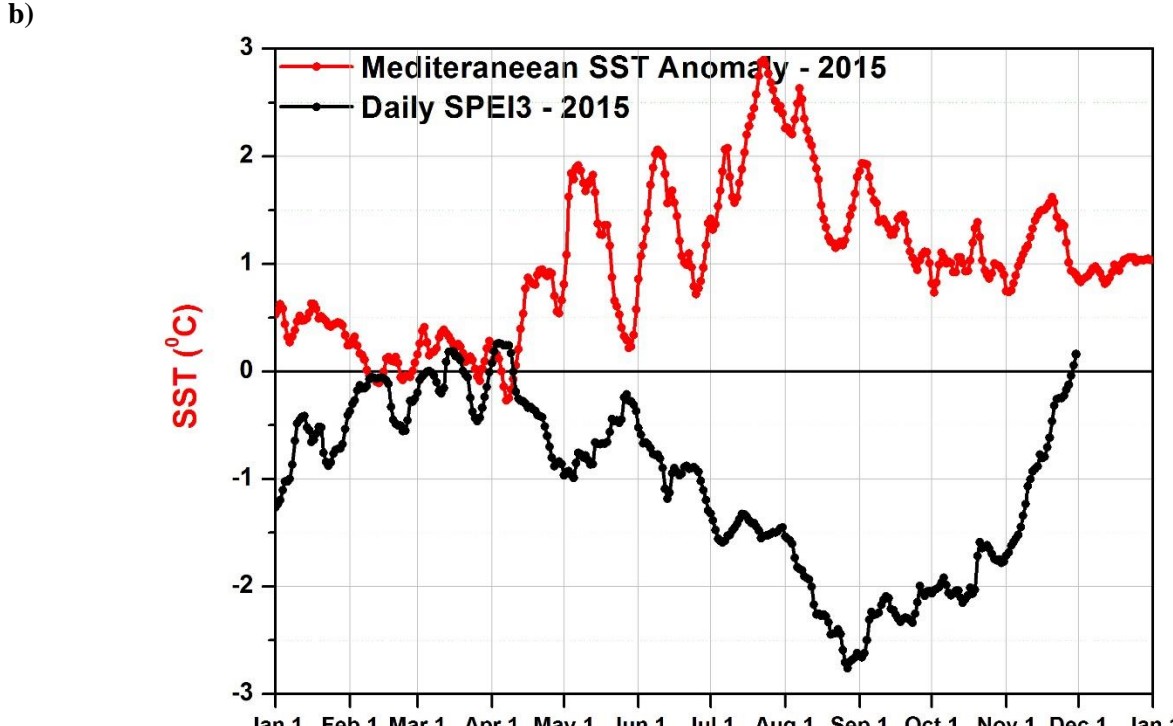

**Figure 5**. a) 2015 Daily SST anomalies computed over the central Atlantic Ocean (blue line) and the Mediterranean Sea (red line); b) 2015 Daily SPEI3 computed over the eastern part of Europe (black line) and the Mediterranean Sea (red line). The daily SST indices are based on the OISST data set [Reynolds at al., 2007].

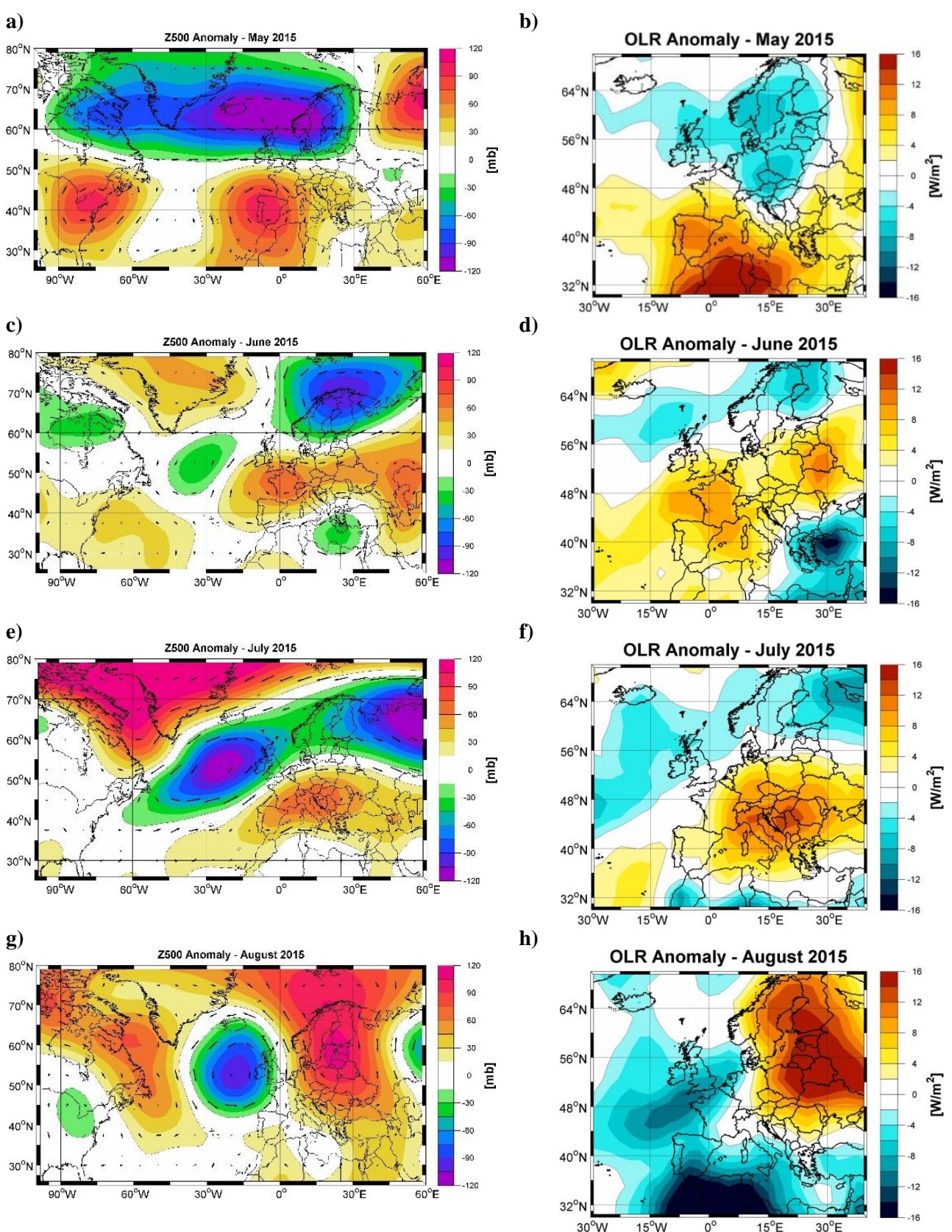

**Figure 6**. Monthly Z500 anomalies (left column) and OLR anomalies (right column) a) and b) May 2015; c) and d) June 2015; e) and f) July 2015; g) and h) August 2015. The anomalies for Z500 and ORL are computed relative to the period 1971 – 2000.

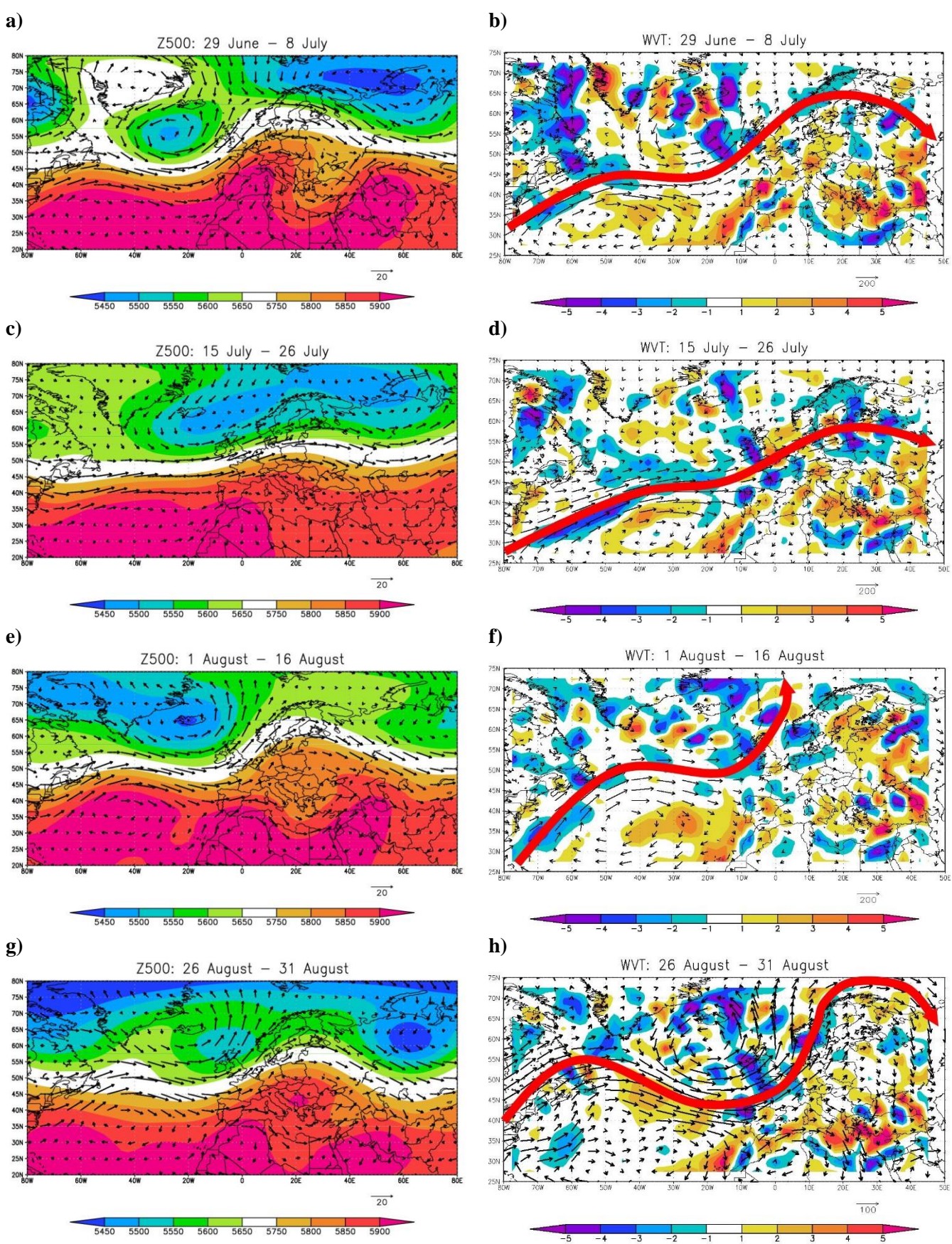

**Figure 7.** Mean daily Z500 (left column) and WVT (right column) averaged over: a) and b) 29 June – 8 July; c) and d) 15 July – 26 July; e) and f) 1 August – 16 August and g) and h) 26 August – 31 August. Units: Z500 (mb)

and WVT (kg m/s).

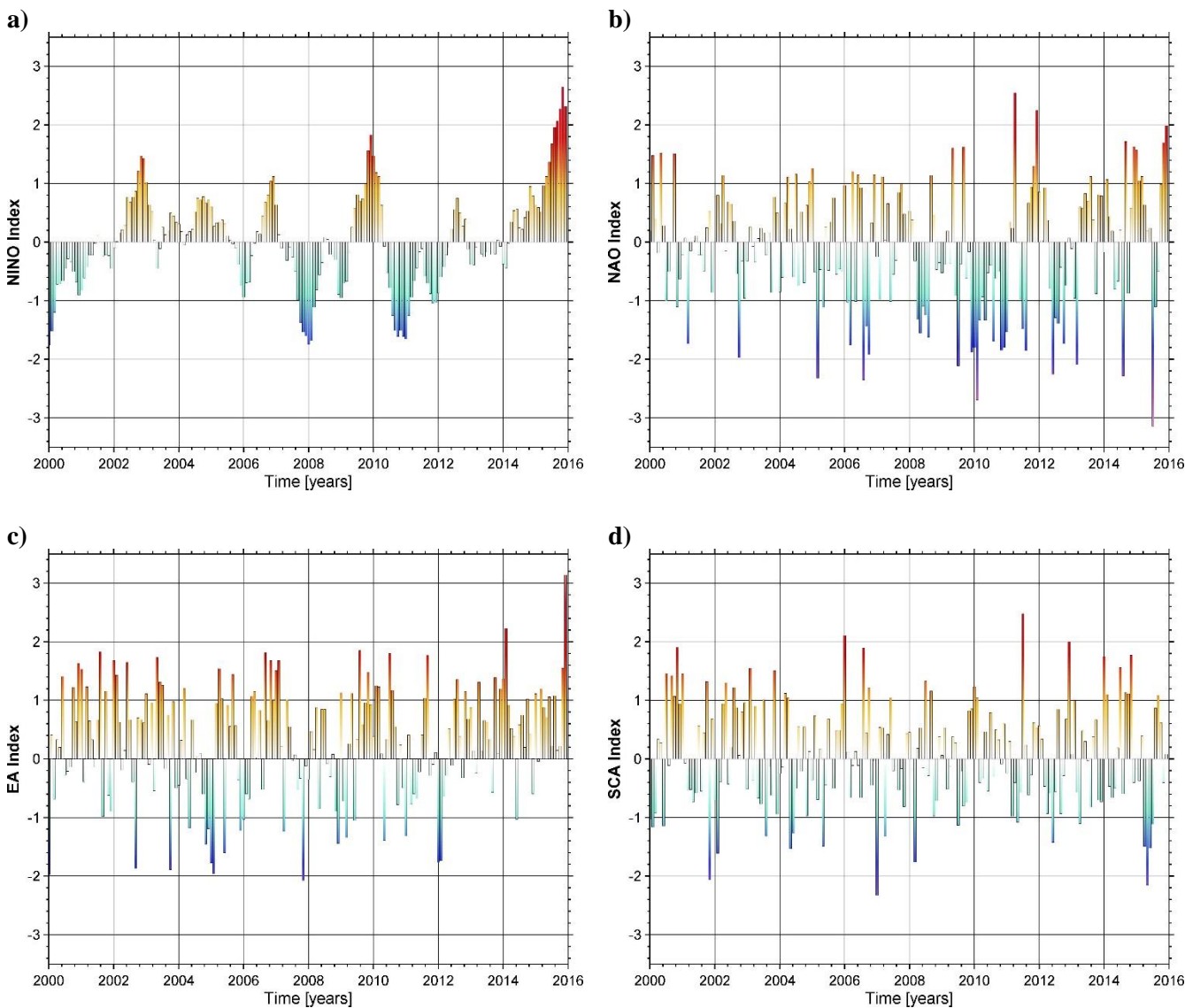

**Figure 8.** Monthly evolution of the teleconnection indices over the period 2006 - 2016: a) Niño 3.4 index; b) NAO index; c) EA index and d) SCA index.

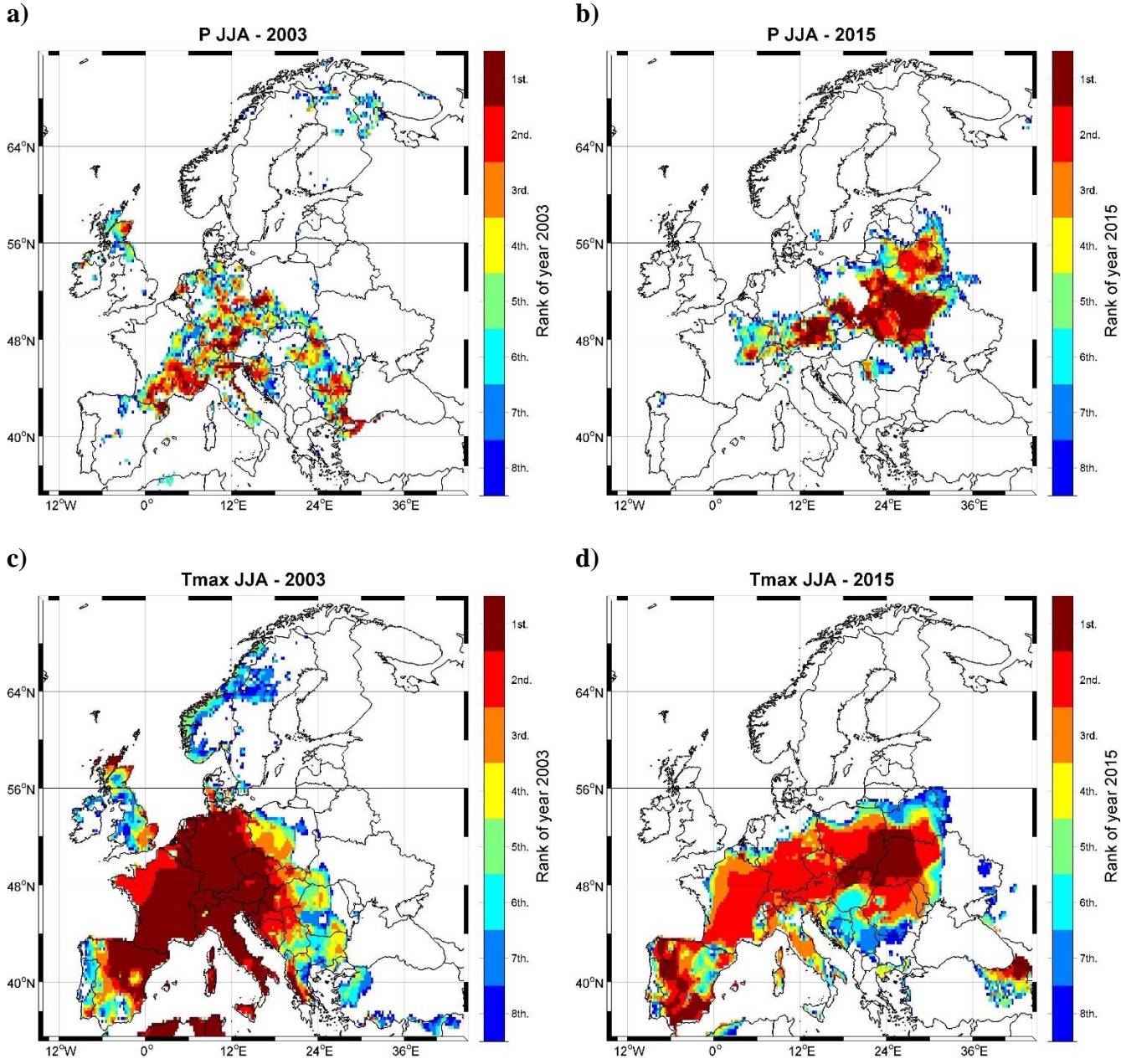

**Figure 9.** a) Top 8 ranking of 2003 summer P; b) Top 8 ranking of 2015 summer PP; c) Top 8 ranking of 2003 summer Tmax and d) Top 8 ranking of 2015 summer Tmax. In this figure, 1 means the driest (P) and warmest (Tmax) summer since 1950, 2 signifies the second driest, and all ranks greater than 8 are shown in white. Analyzed period: 1950 – 2015.

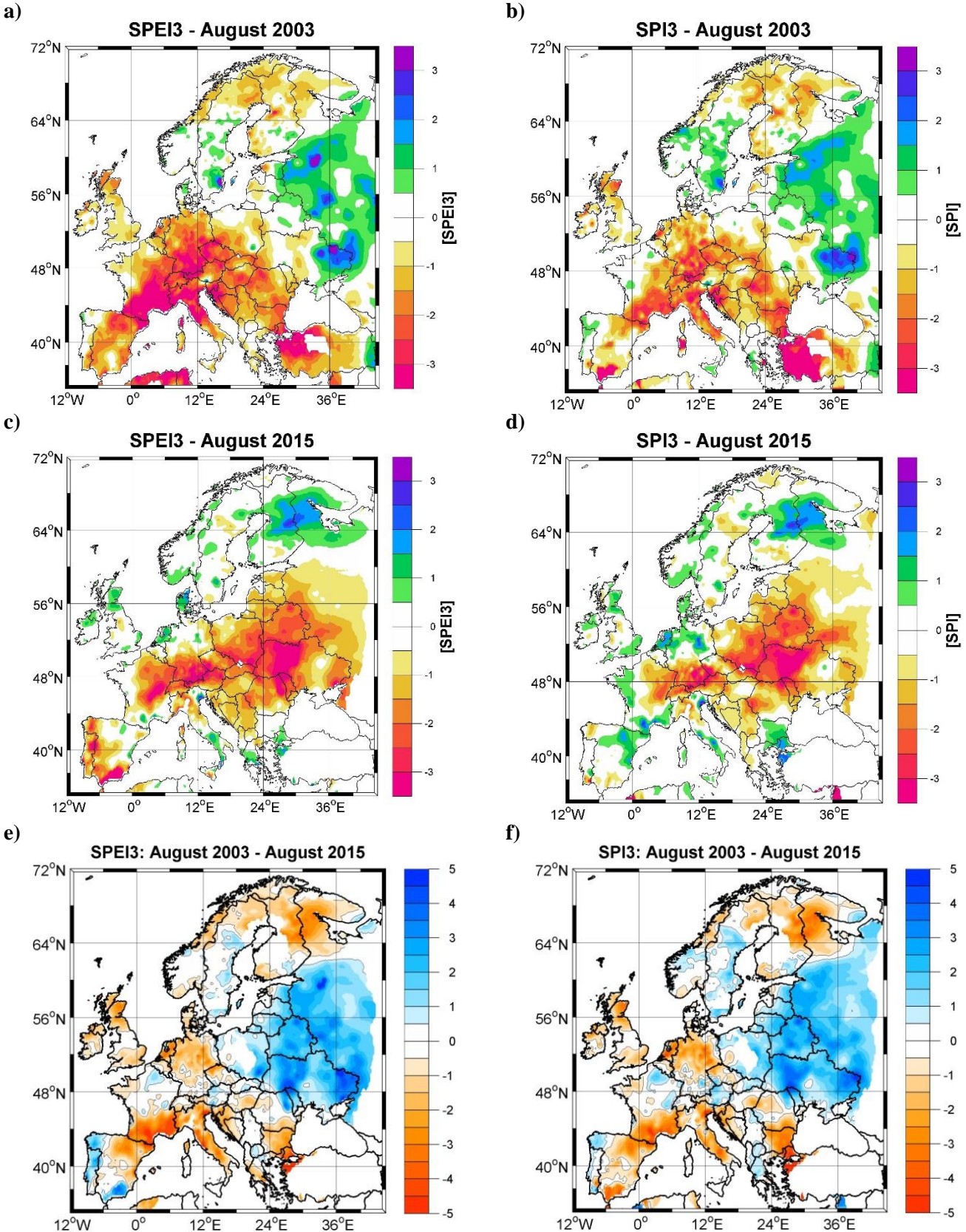

**Figure 10.** a) August 2003 SPEI3; b) as in a) but for SPI3; c) August 2015 SPEI3; d) as in a) but for SPI3; e) The difference between August 2003 SPEI3 and August 2015 SPEI3; f) as in e) but for SPI3.

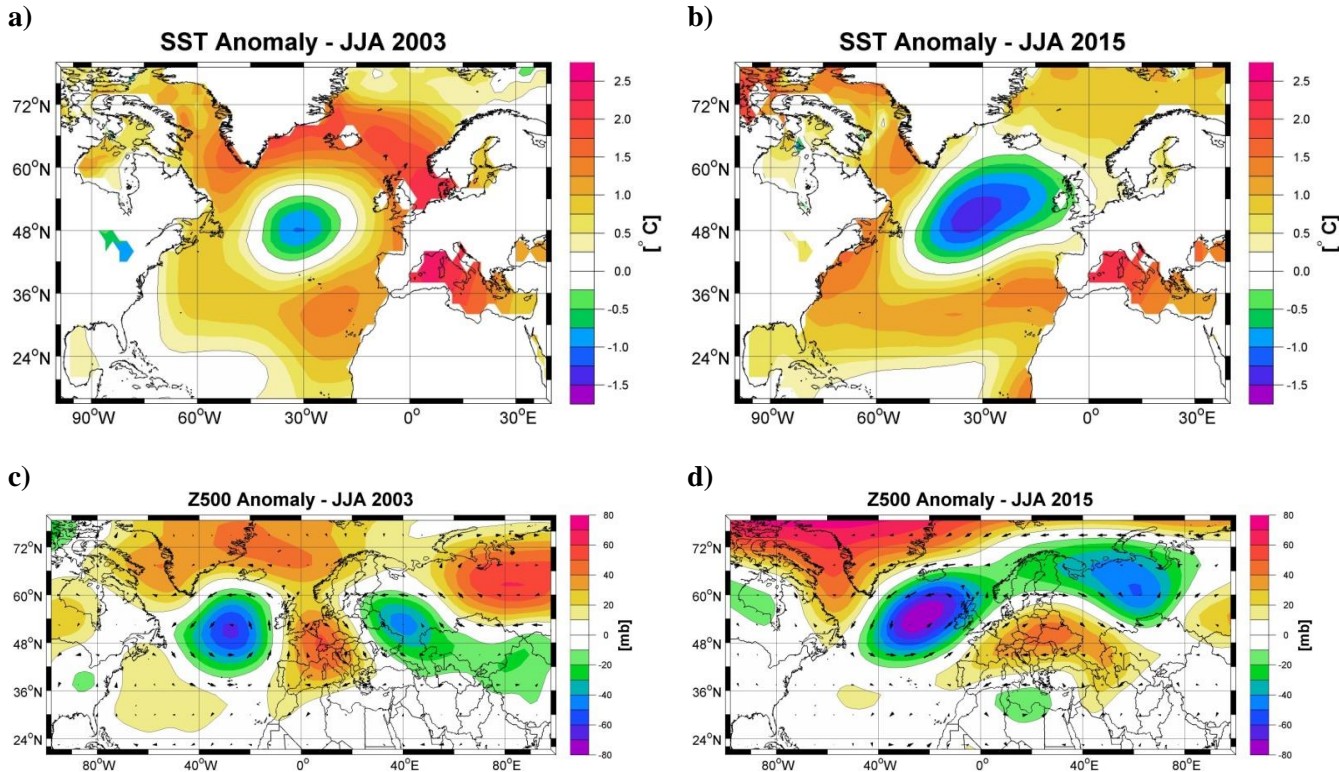

**Figure 11.** a) Summer 2003 SST anomalies; b) as in a) but for summer 2015; c) Summer 2003 Z500 anomalies and d) as in c) but for summer 2015. The anomalies are computed relative to the period 1971 – 2000.

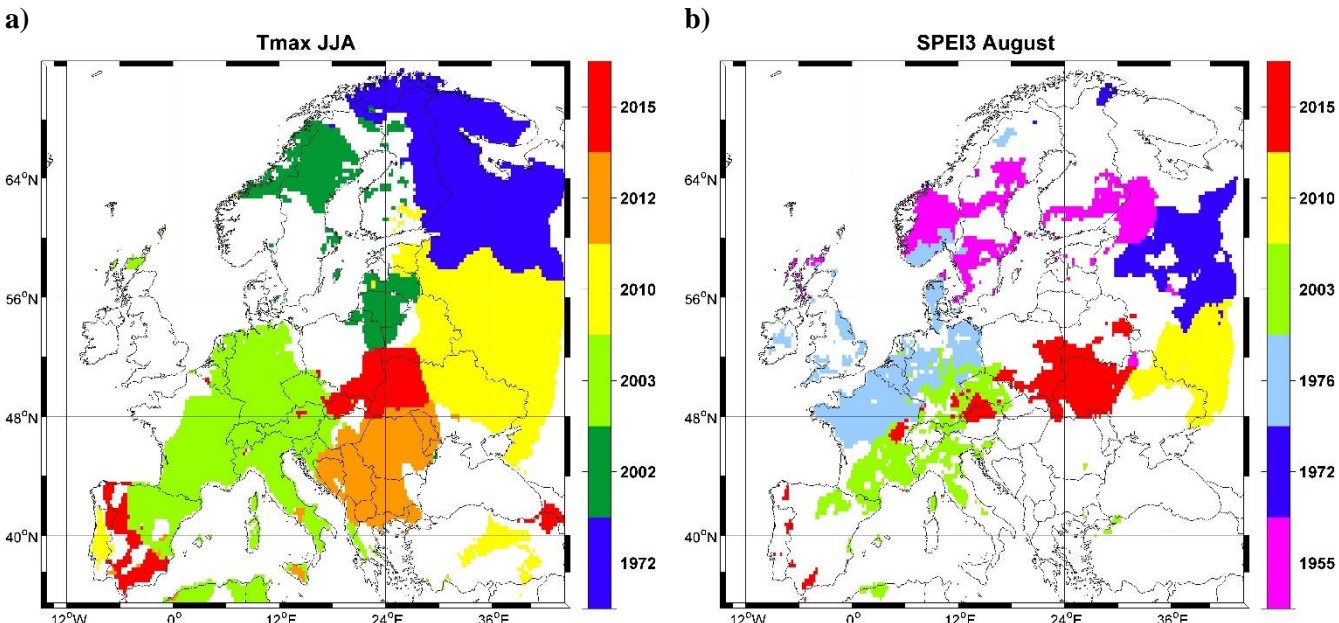

**Figure 12.** a) The spatial extent and the year of record of the warmest summers (Tmax JJA) over the last 66 years over Europe and b) same as in a) but for the driest years (SPEI3). Analyzed period: 1950 – 2015.

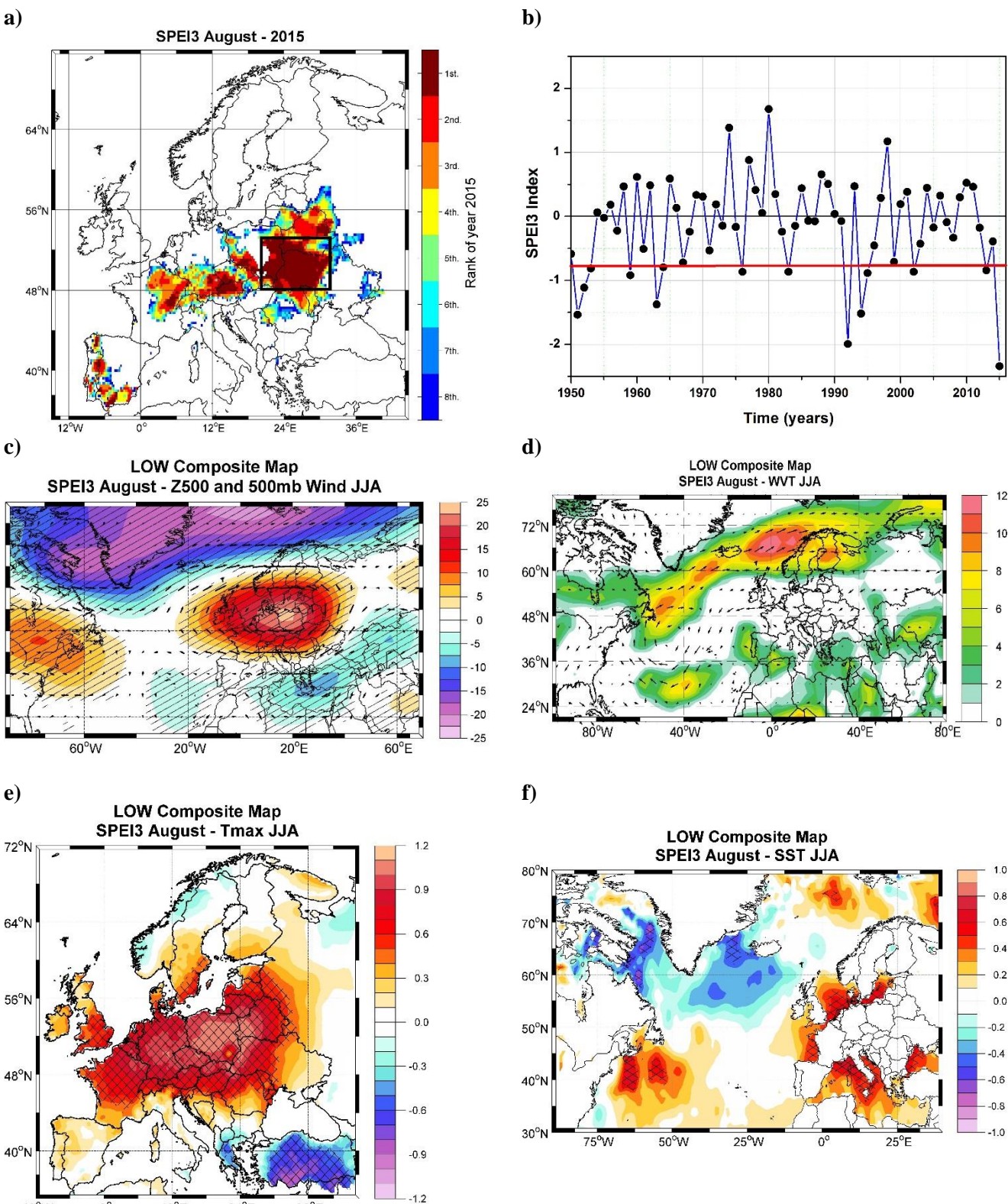

**Figure 13.** a) Top 8 ranking of 2015 August SPEI3; b) The time series of August SPEI3 index averaged over the black box in a); c) The Low (SPEI3 August index ≤ 0.75 standard deviation)) composite map between the August SPEI3 index and the summer (JJA) Z500 and the wind vectors at 500mb; d) the same as c) but for the summer (JJA) WVT; e) the same as c) but for the summer (JJA) Tmax and f) the same as c) but for the summer (JJA) SST. Units: Z500(m), WVT (kg m s$^{-1}$), Tmax (°C) and SST (°C). The hatching highlights significant values at a confidence level of 95%. Analyzed period: 1950 – 2015.

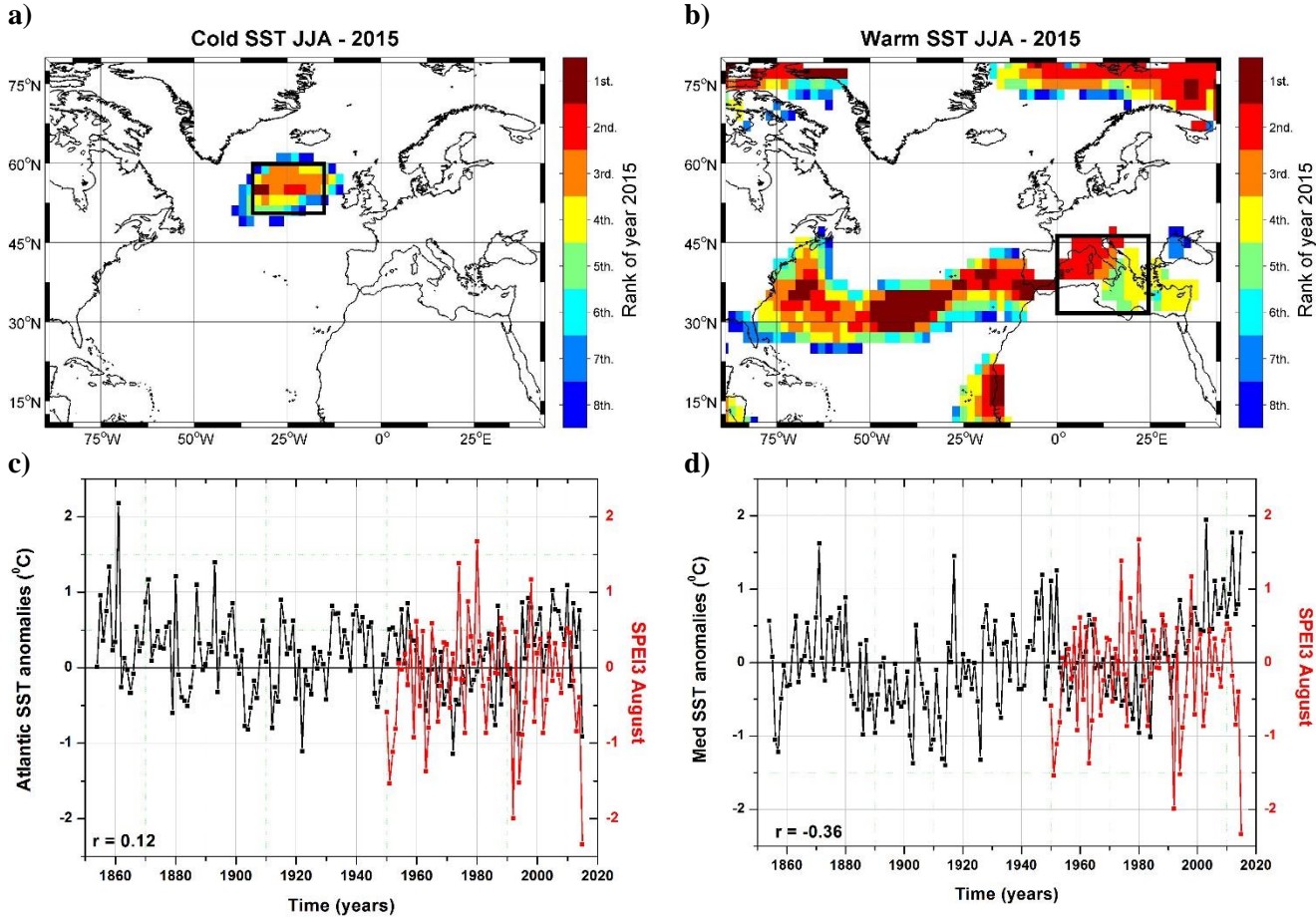

**Figure 14.** a) Top 8 ranking of 2015 summer (JJA) cold SST; b) Top 8 ranking of 2015 summer (JJA) warm SST; c) The time series of JJA cold SST index averaged over the black box in a) (black line) and the time series of August SPEI3 index averaged over the black box in Figure 7a (red line); d) The time series of JJA warm SST index averaged over the black box in b) and the time series of August SPEI3 index averaged over the black box in Figure 7a (red line). The Atlantic SST index and the Med SST index are based on the ERSSTv4b dataset [Liu et al., 2014]. Analyzed period: 1854 – 2015.

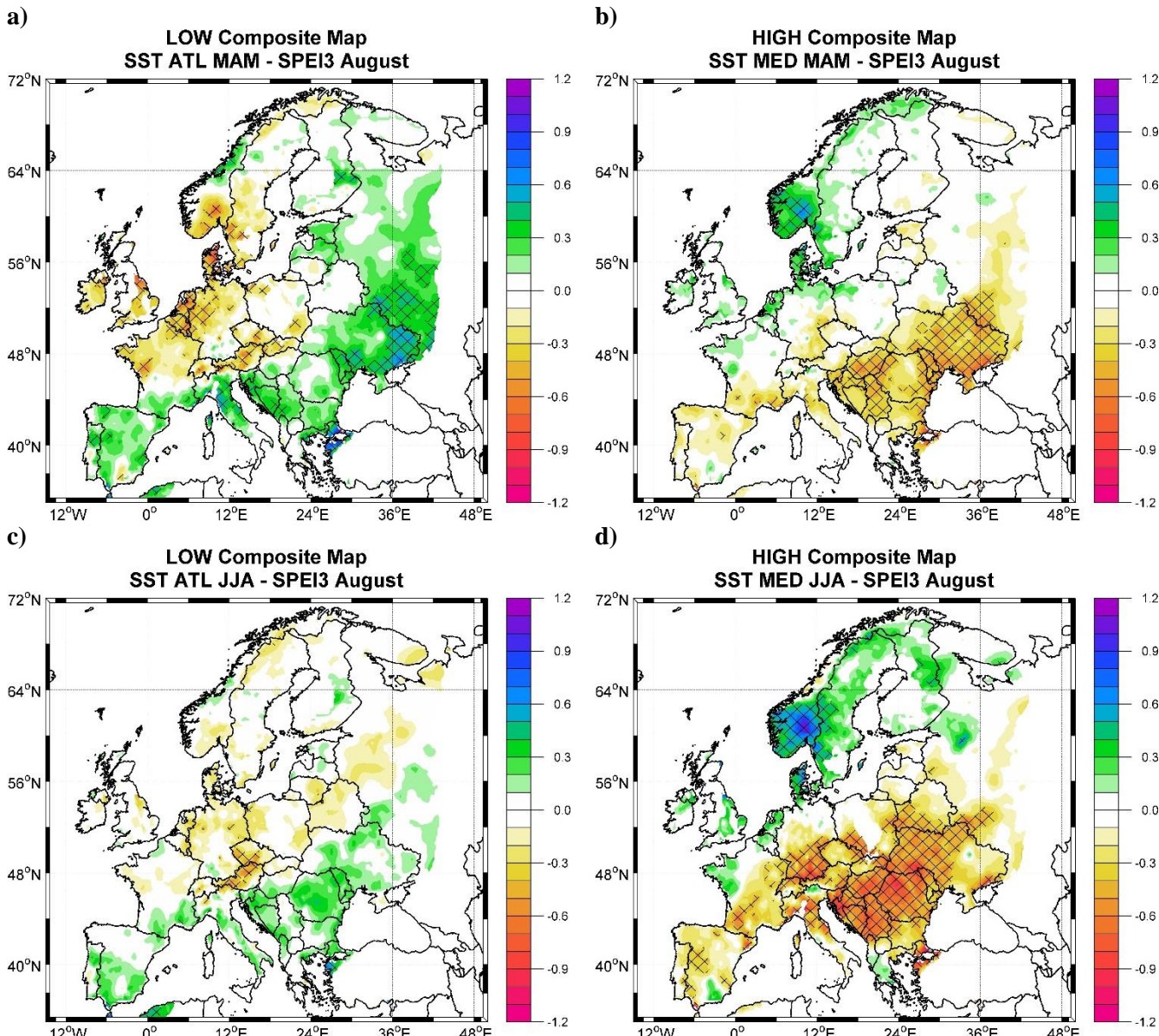

**Figure 15.** a) The Low (**SST cold Index ≤ 0.75 standard deviation**) composite map between the spring (MAM) Atlantic SST cold index (defined as the MAM averaged SST over the black box in Figure 9a) and the August SPEI3 field; b) The High (**SST Warm Index ≥ 0.75 standard deviation**) composite map between the spring (MAM) Mediterranean SST Warm index (defined as the MAM averaged SST over the black box in Figure 9b) and the August SPEI3 field c) same as a) but for summer (JJA) Atlantic SST and d) same as in b) but for the summer (JJA) Mediterranean SST. The hatching highlights significant values at a confidence level of 95%. Analyzed period: 1950 – 2015.