# Peer review of "The European 2015 drought from a climatological perspective"

_Hydrology and Earth System Sciences, 2016_

## Referee Comment (RC1) · M.G.F. Werner (Referee) · 9 Jul 2016

Review:

The European 2015 drought from a climatological perspective

Ionita et al.

This article reviews the European Summer drought of 2015, describing in detail the larger scale climatological characteristics of the drought event, and trying to identify the key drivers that led to the establishing of drought conditions, particularly over the Southern and Eastern Europe. The article is well written and provides a very comprehensive review of the drought event. Additionally a comparison is provided with the 2003 drought event, which showed some distinct differences in spatial extent and initiation, but also similarities with respect to larger scale circulation patterns and the occurrence of anonymously high SST's in the Mediterranean. I think this article, together with its companion paper that studies the 2015 drought event from the hydrological perspective, provide an important insight into the links between the climatological circumstances that lead to drought, the impacts these have on meteorological and hydrological conditions, and the impacts these have on society. I am sure these articles will provide a good reference both to studies that for example explore how climate change may affect the occurrence of drought over Europe, as well as more detailed studies of the 2015 drought and its impacts.

General Comments

While reading the article I was intrigued by the pivotal role of the Mediterranean SST's. One of the objectives of the article is to identify the drivers that lead to the establishing of drought conditions, with the Mediterranean SST's being notes as an important driver. However, the causal relationship is not very clear. I am not a climatologist, so this may be a trivial question, but could the causal relationship be exactly the other way round – i.e. could it be that the warmer SST's in the Mediterranean are the result of the anomalously warm air temperatures. In particular the authors note that in the 2003 event, which started in Spring, the warmer SST's only established themselves in Summer. Also in the discussion, the authors note that the causal effect of the Mediterranean SST's are identified in some studies, but contradicted in others. My question is then if there is more information available from other studies on the causal effect of these warm SST's, or if the reverse causal relationship is possible. I think this is of particular interest to possible anticipation of drought conditions over Southern and Eastern Europe.

Related to this question, the authors have compared the 2015 and 2003 drought events, noting differences in spatial extent but also similarities. While a detailed analysis of other historical drought events would be beyond the scope of this paper, it may be of value if the authors provide any additional information on coincidence of those events with the anomalies in the climatological indicators (e.g. NAO, EA, SCA, Mediterranean SST's) found in 2003 & 2015.

Detailed Comments:

Page 2, Line5: Mention is made of drought impacts of 5000 Billion. I find this number somewhat large. A quick check of the table in the EEA publication reveals this should be about 5 Billion (4.94), or 5000 Million. Please correct (this number is also repeated later in the paper on page 3).

Page 2, Line 15: "*was the warmest on record*" from the context it is implicit that this is globally. To clarify I would add the word "*globally*".

Page 2, line 17: "*air temperature record,* which *were broken*"

Page 2, line 19: "*50 years,* where *only 2003 had lower rainfall*"

Page 3, Line 13: "*precursor* to *dry*"

Page 3, Line 18: "*was* managed *are described*"

Page 4, Line 24: I would rather use "*values* lower", as larger could be confusing.

Page 6, section 3.2: The authors choose to analyse SPI3 and SPEI3. Whilst I agree that this is a good choice given the duration of the events studied, it may be worth commenting on the reason for choosing 3 monthly accumulations, and not 6 or 12 monthly. I would expect this may be relevant for some drought impacts, or even occurrence of hydrological drought (described in the companion paper). Perhaps the authors can add a short note motivation their choice.

Page 7, line 25: It is noted that the SST's were the warmest in 160 years, shown also in Fig 5. What is the reference/source of this time series. This should also be added in section 2.3.

Page 8, line 15: should this not be descending motions?

Page 8: line 19: "throughout *the summer*"

Page 10, line 6: either in "the *western and central part of Europe*" or "Western and Central Europe"

Page 10, line 17: I would rephrase the sentence starting "*In summer 2015*". It is confusing. I would suggest to change to "In 2015, the drought conditions became more evident and accentuated in summer, especially.. ". It is then clearer that the emphasis is on the timing.

Page 10, line 34: "*Mediterranean Sea* alone *could not produce the heat wave*"

Page 12, line 11: "*of the* blocking patterns *over Europe*"

Page 12, line 20: "the *summer 2015* event*, was*"

Page 13, line 1: "The *summer*"

Page 13, line 28: "caution *should be taken*"

Figures:

Figure 1: This figure seems somewhat redundant, and also the figure itself is not very informative. I would expect that the figure would be more relevant to the accompanying paper. Also the discharges of 2015 are compared against the Q80 discharge. It may be more informative to compare against the mean monthly discharges. I would suggest dropping this figure. The reference in the discussion to the accompanying paper should suffice.

Figure 8: While this is included in the Supplementary material, it may be worth extending these figures to 2003 (or 2002) for reference purposes.

Figure 11, caption: "*The anomalies* were *computed*"

---

## Referee Comment (RC2) · Anonymous Referee #2 · 13 Jul 2016

General Comments

I think overall this is a nice climatological overview of the 2015 drought. The comparison with 2003 is also a helpful step in understanding the nature of these droughts (I find the differences in their early development especially interesting), and highlights the challenges of predicting their evolution. While the authors do a good job of outlining the various potential mechanisms that may have led to the drought, the discussion largely reflects our current limited understanding of the causal mechanisms of such droughts, and the limitations of assessing causes from an observational-based study. The weak (if any) link to SST anomalies, the importance of anticyclones, and possible links to various large-scale atmospheric teleconnection patterns, some of which are themselves poorly understood (as well as the potential impact of the overall warming climate) all make understanding the ultimate causes of such droughts a challenge. It

would be nice to see a follow-on modeling study that examined some of the potential causes outlined here in a more quantitative way.

Specific Comments:

While (as the authors note) the well known NAO and SCA patterns appear to play a role in the early and middle stages of the drought, the nature of the blocking pattern that appears to play a key role during August (the warmest month on record) is less clear. In that regard, the authors may find it helpful to take a look at Schubert et. al. (2011) concerning the role of Rossby Waves in summer climate extremes. One of the leading patterns found in that study bears some resemblance to the wave pattern that develops during August 2015.

On a more technical note, I think that since the focus is on the modern era (reference period only goes back to 1971) it might have been better if the authors had used an atmospheric reanalysis that assimilates upper air observations, rather than the 20th century reanalysis, which only assimilates surface pressure. While monthly means are well reproduced in that reanalysis, the results may be less accurate for sub-monthly values. In any event, it might be worth comparing the results in Fig. 7 with e.g. the results based on the older NCAR/NCAR reanalysis just as a sanity check.

Other details:

- please check the cost "5000 billion Euros [EEA, 2010]." – line 5, page 2

- line 18, page 3: "management" should be "managed"

-line 22, page 4: "to" should be "the"

- what is the reference period for the SST anomalies in Fig 5a?

- state the reference period for indices in caption of Fig 8 (1950-2000?)

-page 8 lines 25-28, should note that some studies indicate that the role of the Mediterranean Sea was largely passive in 2003 (e.g., Tomassini and Elizalde 2012)

- "Siegfried et al. 2014" reference should be "Schubert et al. 2014".

- Figure S5 "As in Figure 9", but for the period 1950 – 2015. should be "As in Figure 8".

"Warm Season Subseasonal Variability and Climate Extremes in the Northern Hemisphere: The Role of Stationary Rossby Waves," Schubert, S., H. Wang, and M. Suarez. J. Climate, 24, 4773-4792, 2011.

---

## Referee Comment (RC3) · Anonymous Referee #3 · 22 Jul 2016

General:

An in-depth study of the 2015 European drought would be a valuable addition to the literature. This is clearly a strong group of researchers, but I was very underwhelmed by this article. The analysis is almost entirely descriptive and largely consists of maps of precipitation and temperature anomalies, along with accompanying maps of drought indices. All of this information is easily or widely available. Most of the conclusions reached are obvious: the drought was associated with below average precipitation, above average temperature, positive 500Pa height anomalies, and "widespread areas of negative SPI and SPEI" (quote from Abstract). What drought doesn't have these features? The result that is most interesting is the associated sea-surface temperature patterns, but, like the other analyses, this is a largely descriptive exercise. The comparison with 2003 provides some additional substance, but it makes one wonder about

the features of other past droughts. Why use just one year for the comparison when an ensemble approach is much more valuable? Similarly, no probabilistic information on how unusual (extreme) the 2015 drought was is provided other than that inferred from the SPI and SPEI values. The article reads like a routine government report rather than cutting-edge research.

Specific:

(1) At a minimum, this article needs to incorporate a more quantitative and probabilistic perspective on the 2015 drought. The SPI and SPEI values are a start but don't fully show how unusual the values are. This could be done at each grid point or regionally over appropriate areas (such as agricultural regions or drainage basins).

(2) Building on the lack of probabilistic information, there also is an opportunity to include a paleoclimatic drought perspective. This could easily be done by using data from the Old World Drought Atlas (Cook et al., 2015) and then using that information in a more probabilistic approach.

(3) In terms of what other droughts have occurred and how they compare to these two, the current analysis suggests that the SST dipole may be an important and presumably causal feature. But it is unclear how often this occurs and how long it persists. Is it necessary but not sufficient? What other SST patterns cause extreme droughts in this area? Some additional analysis of that feature could make this a much more useful piece.

Technical:

(1) In the title, "2015 European drought" seems more appropriate than "European 2015 drought". (2) The information on losses of 5000 billion Euro is given twice in the Introduction. (3) The rainbow color map used in Fig. 7 is not appropriate.

References:

Cook, Edward R., et al. "Old World megadroughts and pluvials during the Common

[Figure]

Era." Science Advances 1.10 (2015): e1500561.

---

## Author Comment (AC1) · 22 Aug 2016

We would like to thank the reviewer for the positive feedback on our manuscript and we are grateful for the comments on how it can be further improved. Here, we respond to each comment in turn – full details of the implementation will be provided in the revised manuscript.

Reviewer1

The European 2015 drought from a climatological perspective, by Ionita et al.

This article reviews the European Summer drought of 2015, describing in detail the larger scale climatological characteristics of the drought event, and trying to identify the key drivers that led to the establishing of drought conditions, particularly over the

[Figure]

Southern and Eastern Europe. The article is well written and provides a very comprehensive review of the drought event. Additionally, a comparison is provided with the 2003 drought event, which showed some distinct differences in spatial extent and initiation, but also similarities with respect to larger scale circulation patterns and the occurrence of anonymously high SST's in the Mediterranean. I think this article, together with its companion paper that studies the 2015 drought event from the hydrological perspective, provide an important insight into the links between the climatological circumstances that lead to drought, the impacts these have on meteorological and hydrological conditions, and the impacts these have on society. I am sure these articles will provide a good reference both to studies that for example explore how climate change may affect the occurrence of drought over Europe, as well as more detailed studies of the 2015 drought and its impacts.

General Comments While reading the article I was intrigued by the pivotal role of the Mediterranean SST's. One of the objectives of the article is to identify the drivers that lead to the establishing of drought conditions, with the Mediterranean SST's being notes as an important driver. However, the causal relationship is not very clear. I am not a climatologist, so this may be a trivial question, but could the causal relationship be exactly the other way round – i.e. could it be that the warmer SST's in the Mediterranean is the result of the anomalously warm air temperatures. In particular, the authors note that in the 2003 event, which started in spring, the warmer SST's only established themselves in Summer. Also in the discussion, the authors note that the causal effect of the Mediterranean SST's are identified in some studies, but contradicted in others. My question is then if there is more information available from other studies on the causal effect of these warm SST's, or if the reverse causal relationship is possible. I think this is of particular interest to possible anticipation of drought conditions over Southern and Eastern Europe.

Response: We agree with this comment, i.e. the causal relationship is not clear, and it is not obvious what is driving what. In the manuscript, we have tried to emphasize that

the Mediterranean SST does have a role in influencing the heat waves and droughts over Europe, but the real mechanism behind this relationship is not fully understood. There are some modelling studies (see references in the manuscript) that have tried to deal with this issue, but the results are not entirely conclusive or they are contradictory. From an observational point of view, the role of the Atlantic and Mediterranean SST and droughts, at European level, has been analyzed in various papers (see the references in the manuscript), but without model simulations and sensitivity studies it is rather difficult to have a clear picture of this relationship. The causality would require a complex model analysis that considers various factors. This is also commented on by Referee #2, highlighting the limitations of assessing causes from an observational-based study and the recommendations to make a follow-on modelling study. We will revisit the text with the aim to make these considerations clearer to the reader.

Related to this question, the authors have compared the 2015 and 2003 drought events, noting differences in spatial extent but also similarities. While a detailed analysis of other historical drought events would be beyond the scope of this paper, it may be of value if the authors provide any additional information on coincidence of those events with the anomalies in the climatological indicators (e.g. NAO, EA, SCA, Mediterranean SST's) found in 2003 & 2015.

Response: In the revised version of the manuscript, we will add some information regarding the influence of different teleconnection patterns on other drought events.

Detailed Comments: Page 2, Line5: Mention is made of drought impacts of 5000 Billion. I find this number somewhat large. A quick check of the table in the EEA publication reveals this should be about 5 Billion (4.94), or 5000 Million. Please correct (this number is also repeated later in the paper on page 3).

Response: Thank you for noting this error (will be corrected).

Page 2, Line 15: "was the warmest on record" from the context it is implicit that this is globally. To clarify I would add the word "globally".

Response: The text will be modified accordingly.

Page 2, line 17: "air temperature record, which were broken" Page 2, line 19: "50 years, where only 2003 had lower rainfall" Page 3, Line 13: "precursor to dry" Page 3, Line 18: "was managed are described" Page 4, Line 24: I would rather use "values lower", as larger could be confusing.

Response: All of the above suggestions will be inserted in the revised version of the manuscript.

Page 6, section 3.2: The authors choose to analyse SPI3 and SPEI3. Whilst I agree that this is a good choice given the duration of the events studied, it may be worth commenting on the reason for choosing 3 monthly accumulations, and not 6 or 12 monthly. I would expect this may be relevant for some drought impacts, or even occurrence of hydrological drought (described in the companion paper). Perhaps the authors can add a short note motivation their choice.

Response: Thank you for highlighting this method question. We will add a sentence better supporting our decision to use a 3-month anomaly. In prior work (unpublished) leading up to Kingston et al. (2015), our team found only minimal differences between the atmospheric drivers for 3 months and 6-month SPI/SPEI drought events. In addition, a 3-month anomaly has been shown to be important for a broad range of drought-related impacts, including agricultural losses, water supply, and freshwater ecosystem impacts (Stagge et al., 2015 - Modeled drought impact occurrence based on meteorological drought indices in Europe). In this case, a 3-months accumulation period was seen as a practical compromise used in monitoring to capture seasonal patterns and to estimate many important drought-related impacts.

Page 7, line 25: It is noted that the SST's were the warmest in 160 years, shown also in Fig 5. What is the reference/source of this time series. This should also be added in section 2.3. Page 8, line 15: should this not be descending motions? Page 8: line 19: "throughout the summer" Page 10, line 6: either in "the western and central part
of Europe" or "Western and Central Europe" Page 10, line 17: I would rephrase the sentence starting "In summer 2015". It is confusing. I would suggest to change to "In 2015, the drought conditions became more evident and accentuated in summer, especially.. ". It is then clearer that the emphasis is on the timing. Page 10, line 34: "Mediterranean Sea alone could not produce the heat wave" Page 12, line 11: "of the blocking patterns over Europe" Page 12, line 20: "the summer 2015 event, was" Page 13, line 1: "The summer" Page 13, line 28: "caution should be taken"

Response: All the above suggestions/comments will be inserted in the revised version of the manuscript.

Figures: Figure 1: This figure seems somewhat redundant, and also the figure itself is not very informative. I would expect that the figure would be more relevant to the accompanying paper. Also the discharges of 2015 are compared against the Q80 discharge. It may be more informative to compare against the mean monthly discharges. I would suggest dropping this figure. The reference in the discussion to the accompanying paper should suffice.

Response: We agree with this comment. As such, we will remove Figure 1 in the revised version of the manuscript.

Figure 8: While this is included in the Supplementary material, it may be worth extending these figures to 2003 (or 2002) for reference purposes.

Response: We will extend the figures according to the suggestion.

Figure 11, caption: "The anomalies were computed"

Response: The text will be modified accordingly.

---

## Author Comment (AC2) · 22 Aug 2016

We would like to thank the reviewer for the positive feedback on our manuscript and we are grateful for the comments on how it can be further improved. Here, we respond to each comment in turn – full details of the implementation will be provided in the revised manuscript.
I think overall this is a nice climatological overview of the 2015 drought. The comparison with 2003 is also a helpful step in understanding the nature of these droughts (I

find the differences in their early development especially interesting), and highlights the challenges of predicting their evolution. While the authors do a good job of outlining the various potential mechanisms that may have led to the drought, the discussion largely reflects our current limited understanding of the causal mechanisms of such droughts, and the limitations of assessing causes from an observational-based study.

The weak (if any) link to SST anomalies, the importance of anticyclones, and possible links to various large-scale atmospheric teleconnection patterns, some of which are themselves poorly understood (as well as the potential impact of the overall warming climate) all make understanding the ultimate causes of such droughts a challenge. It would be nice to see a follow-on modeling study that examined some of the potential causes outlined here in a more quantitative way.

Response: We agree with this comment, and as highlighted in our response to Referee #1 we have tried to emphasize also in the manuscript the fact that the Mediterranean SST does have a role in influencing heat waves and droughts over Europe, but the real mechanism behind this relationship is not fully understood. The causality would require a complex model analysis that considers various factors. Nevertheless, we do anticipate designing a sensitivity experiment using a coupled atmosphere-ocean model such as (V. Artale et al.: An atmosphere–ocean regional climate model for the Mediterranean area) by increasing the SST in the Mediterranean region and observing the response of the atmosphere to this increase. This is beyond the scope of the present paper, but we hope that this manuscript will be a starting point for such modelling analyses.

Specific Comments: While (as the authors note) the well-known NAO and SCA patterns appear to play a role in the early and middle stages of the drought, the nature of the blocking pattern that appears to play a key role during August (the warmest month on record) is less clear. In that regard, the authors may find it helpful to take a look at Schubert et. al. (2011) concerning the role of Rossby Waves in summer climate extremes. One of the leading patterns found in that study bears some resemblance to the wave pattern that develops during August 2015.

Response: We thank the reviewer for this valuable comment and we will add some more information in the revised version of the manuscript regarding the influence of the blocking pattern on the development of the drought event.

On a more technical note, I think that since the focus is on the modern era (reference period only goes back to 1971) it might have been better if the authors had used an atmospheric reanalysis that assimilates upper air observations, rather than the 20th century reanalysis, which only assimilates surface pressure. While monthly means are well reproduced in that reanalysis, the results may be less accurate for sub-monthly values. In any event, it might be worth comparing the results in Fig. 7 with e.g. the results based on the older NCAR/NCAR reanalysis just as a sanity check.

Response: We apologize for the confusion. The reference to the data sets in our manuscript is misleading. We have actually used the NCEP/NCAR Reanalysis 1 (1948 – 2015). We started our analysis using the 20th century reanalysis, but realizing similar concerns to the ones you mention, all the results and figures in the paper have been obtained based on the NCEP/NCAR Reanalysis 1 (1948 – 2015). We missed this change to the references and have modified the revised manuscript to add the proper set of data.

Other details: - please check the cost "5000 billion Euros [EEA, 2010]." – line 5, page 2 - line 18, page 3: "management" should be "managed" - line 22, page 4: "to" should be "the" - what is the reference period for the SST anomalies in Fig 5a? - state the reference period for indices in caption of Fig 8 (1950-2000?) - page 8 lines 25-28, should note that some studies indicate that the role of the Mediterranean Sea was largely passive in 2003 (e.g., Tomassini and Elizalde 2012) - "Siegfried et al. 2014" reference should be "Schubert et al. 2014". - Figure S5 "As in Figure 9", but for the period 1950 – 2015. should be "As in Figure 8".

Response: All the above suggestions/comments will be accounted for in the revised version of the manuscript.

"Warm Season Subseasonal Variability and Climate Extremes in the Northern Hemisphere: The Role of Stationary RossbyWaves," Schubert, S., H.Wang, and M. Suarez. J. Climate, 24, 4773-4792, 2011

---

## Author Comment (AC3) · 22 Aug 2016

We would like to thank the reviewer for the constructive feedback on our manuscript and we are grateful for the comments on how it can be further improved. Here, we respond to each comment in turn – full details of the implementation will be provided in the revised manuscript.
General: An in-depth study of the 2015 European drought would be a valuable addition to the literature. This is clearly a strong group of researchers, but I was very

underwhelmed by this article. The analysis is almost entirely descriptive and largely consists of maps of precipitation and temperature anomalies, along with accompanying maps of drought indices. All of this information is easily or widely available. Most of the conclusions reached are obvious: the drought was associated with below average precipitation, above average temperature, positive 500Pa height anomalies, and "widespread areas of negative SPI and SPEI" (quote from Abstract). What drought doesn't have these features? The result that is most interesting is the associated sea-surface temperature patterns, but, like the other analyses, this is a largely descriptive exercise. The comparison with 2003 provides some additional substance, but it makes one wonder about the features of other past droughts. Why use just one year for the comparison when an ensemble approach is much more valuable? Similarly, no probabilistic information on how unusual (extreme) the 2015 drought was is provided other than that inferred from the SPI and SPEI values. The article reads like a routine government report rather than cutting-edge research.

Response: As clarified in our response to Reviewer 1, the aim of our study was to have a broader overview of the drivers of the summer 2015 drought both from a climatological point of view (present paper) as well as from a hydrological point of view (http://www.hydrol-earth-syst-sci-discuss.net/hess-2016-366/), following the pattern of previous atmospheric summaries of individual climatic extreme events. Analyzing and managing drought in a pro-active way requires a concerted action of the hydrological and climatic communities. This twin papers are a first attempt to emphasize the need of such actions from different communities. The hydrological portion is fully covered by the twin paper mentioned before. In this way, the two papers are largely descriptive, but are designed as a clear and comprehensive summary of the 2015 event to promote more detailed future studies by compiling available data. There is a long history of extreme event summaries published in the atmospheric science literature, from 1970s-80s to the yearly special issues of the Bulletin of American Meteorological Society – State of the Climate (https://www.ncdc.noaa.gov/bams). These studies have been designed to highlight the unique features of a given event and to spur/urge/promote more

detailed modeling research. This study follows this tradition, but along with its companion hydrological paper, summarizes the drought event from both, a meteorological and hydrological perspective, focusing on drought indices used operationally and quantifying the larger water cycle effects.

Specific: (1) At a minimum, this article needs to incorporate a more quantitative and probabilistic perspective on the 2015 drought. The SPI and SPEI values are a start but don't fully show how unusual the values are. This could be done at each grid point or regionally over appropriate areas (such as agricultural regions or drainage basins).

Response: To add more quantitative measure to our analysis, we will incorporate ranking maps in the revised manuscript (see Figure S1 at the end of this document), which will show where summer 2015 ranks, in terms of amplitude, compared to the last 65 years. The ranking is done in every grid point.

(2) Building on the lack of probabilistic information, there also is an opportunity to include a paleoclimatic drought perspective. This could easily be done by using data from the Old World Drought Atlas (Cook et al., 2015) and then using that information in a more probabilistic approach.

Response: Including a paleo perspective, while interesting, would be beyond the scope of our paper. Moreover, a comparison between the drought 2015 and the Old World Drought Atlas would not be feasible, due to the different data sets involved. Our drought measures (SPI, SPEI) are based on observed data (e.g. precipitation, temperature) and verified algorithms, while the self-calibrating Palmer Drought Severity Index data from the world drought atlas is based on proxy data (e.g. tree rings). Furthermore, the scPDSI from the Old World Atlas does not cover the year 2015, which would make such a comparison impossible. Finally, and perhaps most critical, is that the regions most affected by drought in 2015 (the eastern part of Poland and Ukraine), are poorly or not at all covered by tree rings sites in the Old World Atlas scPDSI reconstruction (see Figure 1 in Cook et al. 2015), thus making the drought reconstruction over these

regions not reliable.

(3) In terms of what other droughts have occurred and how they compare to these two, the current analysis suggests that the SST dipole may be an important and presumably causal feature. But it is unclear how often this occurs and how long it persists. Is it necessary but not sufficient? What other SST patterns cause extreme droughts in this area? Some additional analysis of that feature could make this a much more useful piece.

Response: More complex analysis (e.g. Empirical Orthogonal Function Analysis (EOF) and Composite Maps Analysis (CMA)) will be employed to tackle these issues and new results based on these statistical measures will be added to the revised manuscript.

Technical: (1) In the title, "2015 European drought" seems more appropriate than "European 2015 drought". (2) The information on losses of 5000 billion Euro is given twice in the Introduction. (3) The rainbow color map used in Fig. 7 is not appropriate.

Response: All these technical issues will be accounted for in the revised version of the manuscript.

References: Cook, Edward R., et al. "Old World megadroughts and pluvials during the Common Era." Science Advances 1.10 (2015): e1500561.
* * *
* * *
[Figure]

*Figure S1.* Top 8 ranking of summer (JJA) SPEI3 severity for the (a) 2003 and (b) 2015 drought events. In this figure, 1 means the driest summer since 1950, 2 signifies the second driest, and all ranks greater than 8 are shown in white. Tmax for summer (JJA) are shown in a similar manner for the (c) 2003 and (d) 2015 drought events.

**Fig. 1.**

---

## Author Response (AR1)

**Dear Dr. Stahl,**

We appreciate the feedback that we received from the three reviewers. In response to the reviewers concerns, we submit a revised version of the paper, in which we took into account all their comments/suggestions. We have in particular extended our analysis on the influence of the Mediterranean and North Atlantic Ocean SST on the occurrence of summer droughts in Europe (causal relationship), and further included a new type of analysis (rank maps), thus being able to place the 2015 event in a long term context. The changes that have been made have improved both the scope and delivery of the paper. Moreover, we have considered some other new studies that have been published regarding the 2015 event and emphasized the added value of our analysis compared to these studies.

We submit, together with the revised version of the manuscript, a separate response to all the referees' comments describing the specific changes and additions that we have made.
We feel that this research is of interest to the general readership of HESS as well as to a larger public audience, and will also appeal to specialists in the fields of climate modeling, climatology, hydrology and water management.
All authors have agreed with resubmission of this revised manuscript, and no part of the paper is published or under review at another journal.

Sincerely,

Monica Ionita

We would like to thank the reviewers for their positive feedback on our manuscript and we are grateful for the comments on how it can be further improved. We provide below a point by point response to the reviewers comments/suggestions as well as the modified version of the manuscript (end of the document) with the changes highlighted in red.

**Reviewer1**

**The European 2015 drought from a climatological perspective, by Ionita et al.**

This article reviews the European Summer drought of 2015, describing in detail the larger scale climatological characteristics of the drought event, and trying to identify the key drivers that led to the establishing of drought conditions, particularly over the Southern and Eastern Europe. The article is well written and provides a very comprehensive review of the drought event. Additionally, a comparison is provided with the 2003 drought event, which showed some distinct differences in spatial extent and initiation, but also similarities with respect to larger scale circulation patterns and the occurrence of anonymously high SST's in the Mediterranean. I think this article, together with its companion paper that studies the 2015 drought event from the hydrological perspective, provide an important insight into the links between the climatological circumstances that lead to drought, the impacts these have on meteorological and hydrological conditions, and the impacts these have on society. I am sure these articles will provide a good reference both to studies that for example explore how climate change may affect the occurrence of drought over Europe, as well as more detailed studies of the 2015 drought and its impacts.

**General Comments**

While reading the article I was intrigued by the pivotal role of the Mediterranean SST's. One of the objectives of the article is to identify the drivers that lead to the establishing of drought conditions, with the Mediterranean SST's being notes as an important driver. However, the causal relationship is not very clear. I am not a climatologist, so this may be a trivial question, but could the causal relationship be exactly the other way round – i.e. could it be that the warmer SST's in the Mediterranean is the result of the anomalously warm air temperatures. In particular, the authors note that in the 2003 event, which started in spring, the warmer SST's only established themselves in Summer. Also in the discussion, the authors note that the causal effect of the Mediterranean SST's are identified in some studies, but contradicted in others. My question is then if there is more information available from other studies on the causal effect of these warm SST's, or if the reverse causal relationship is possible. I think this is of particular interest to possible anticipation of drought conditions over Southern and Eastern Europe.

**Response:** In the revised version of the manuscript we have tried to expand the analysis and discussion regarding the influence of the Mediterranean SST on the occurrence of summer droughts over the European region. We have extended our analysis over the last 66 years (please see Section 4.1 and the new Section 4.4) and shown preliminary tests to determine causal relationships with SST. Nevertheless, a full picture of the causal relationship between Mediterranean SST and drought condition would require more complex and targeted model analysis that is beyond the scope of this paper, which is already quite long and detailed.

Related to this question, the authors have compared the 2015 and 2003 drought events, noting differences in spatial extent but also similarities. While a detailed analysis of other historical drought events would

be beyond the scope of this paper, it may be of value if the authors provide any additional information on coincidence of those events with the anomalies in the climatological indicators (e.g. NAO, EA, SCA, Mediterranean SST's) found in 2003 & 2015.

**Response:** Following the aforementioned suggestion, we have modified the text accordingly – Section 4.3.

Detailed Comments:
Page 2, Line5: Mention is made of drought impacts of 5000 Billion. I find this number somewhat large. A quick check of the table in the EEA publication reveals this should be about 5 Billion (4.94), or 5000 Million. Please correct (this number is also repeated later in the paper on page 3).

**Response:** The text has been modified with the proper number (page 2, lines 5 – 7).

Page 2, Line 15: "*was the warmest on record*" from the context it is implicit that this is globally. To clarify I would add the word "*globally*".

**Response:** The text has been modified accordingly.

Page 2, line 17: "*air temperature record,* which *were broken*"
Page 2, line 19: "*50 years,* where *only 2003 had lower rainfall*"
Page 3, Line 13: "*precursor* to *dry*"
Page 3, Line 18: "*was* managed *are described*"
Page 4, Line 24: I would rather use "*values* lower", as larger could be confusing.

**Response:** All of the above suggestions have been inserted in the revised version of the manuscript.

Page 6, section 3.2: The authors choose to analyze SPI3 and SPEI3. Whilst I agree that this is a good choice given the duration of the events studied, it may be worth commenting on the reason for choosing 3 monthly accumulations, and not 6 or 12 monthly. I would expect this may be relevant for some drought impacts, or even occurrence of hydrological drought (described in the companion paper). Perhaps the authors can add a short note motivation their choice.

**Response:** Thank you for highlighting this method question. We have added new text to better support our decision to use a 3-month anomaly (page 5, lines 23 – 26).

Page 7, line 25: It is noted that the SST's were the warmest in 160 years, shown also in Fig 5. What is the reference/source of this time series. This should also be added in section 2.3.
Page 8, line 15: should this not be descending motions?
Page 8: line 19: "throughout *the summer*"
Page 10, line 6: either in "the *western and central part of Europe*" or "Western and Central Europe"
Page 10, line 17: I would rephrase the sentence starting "*In summer 2015*". It is confusing. I would suggest to change to "In 2015, the drought conditions became more evident and accentuated in summer, especially.. ". It is then clearer that the emphasis is on the timing.
Page 10, line 34: "*Mediterranean Sea* alone *could not produce the heat wave*"
Page 12, line 11: "*of the* blocking patterns *over Europe*"
Page 12, line 20: "the *summer 2015* event*, was*"

Page 13, line 1: "The *summer*"
Page 13, line 28: "caution *should be taken*"

**Response:** All the above suggestions/comments have been inserted in the revised version of the manuscript.

Figures:
Figure 1: This figure seems somewhat redundant, and also the figure itself is not very informative. I would expect that the figure would be more relevant to the accompanying paper. Also the discharges of 2015 are compared against the Q80 discharge. It may be more informative to compare against the mean monthly discharges. I would suggest dropping this figure. The reference in the discussion to the accompanying paper should suffice.

**Response:** We agree with this comment. As such, we have removed Figure 1 from the revised version of the manuscript.

Figure 8: While this is included in the Supplementary material, it may be worth extending these figures to 2003 (or 2002) for reference purposes.

**Response:** In the new version of the manuscript we have inserted new figures and tried to improve the existing figures based on reviewer recommendations. Figure 8 has been extended over the period 2000 – 2015 (see Figure 8 in the revised manuscript).

Figure 11, caption: "*The anomalies* were *computed*"

**Response:** The text has been modified accordingly.

**Anonymous Referee #2**

General Comments

I think overall this is a nice climatological overview of the 2015 drought. The comparison with 2003 is also a helpful step in understanding the nature of these droughts (I find the differences in their early development especially interesting), and highlights the challenges of predicting their evolution. While the authors do a good job of outlining the various potential mechanisms that may have led to the drought, the discussion largely reflects our current limited understanding of the causal mechanisms of such droughts, and the limitations of assessing causes from an observational-based study.

The weak (if any) link to SST anomalies, the importance of anticyclones, and possible links to various large-scale atmospheric teleconnection patterns, some of which are themselves poorly understood (as well as the potential impact of the overall warming climate) all make understanding the ultimate causes of such droughts a challenge. It would be nice to see a follow-on modeling study that examined some of the potential causes outlined here in a more quantitative way.

**Response:** We appreciate this comment and agree that more study is needed to fully characterize the drivers of European droughts. We do anticipate a future "follow-on modeling study", designed as a sensitivity experiment using a coupled atmosphere-ocean model such as V. Artale et al.: (An atmosphere–ocean regional climate model for the Mediterranean area) to increase the SST in the Mediterranean region and observe the response of the atmosphere to this increase. It is our hope that this manuscript will be a starting point for such a study, but we believe this is beyond the scope of the present paper.

In order to further support future modeling studies, we have added more detailed statistical analysis regarding the influence of the Mediterranean SST on the occurrence of summer droughts over the European region. We have extended our analysis over the last 66 years (please see Sections 4.1 and 4.4) and shown that warmer Mediterranean SST have coincided with drought events in the historical record in a manner similar to the 2015 event, while the SST anomaly in the North Atlantic does not have a historical link. As explained above, these observations must be validated through atmosphere-ocean modeling, but we hope that our additional analysis provides even stronger motivation to pursue this research in the future.

Specific Comments:
While (as the authors note) the well-known NAO and SCA patterns appear to play a role in the early and middle stages of the drought, the nature of the blocking pattern that appears to play a key role during August (the warmest month on record) is less clear. In that regard, the authors may find it helpful to take a look at Schubert et. al. (2011) concerning the role of Rossby Waves in summer climate extremes. One of the leading patterns found in that study bears some resemblance to the wave pattern that develops during August 2015.

**Response:** We thank the reviewer for this valuable comment. We added some more information in the revised version of the manuscript regarding the influence of the blocking pattern on the development of the drought events (pages 10, lines 29 – 32; page 11, lines 15 -19).

On a more technical note, I think that since the focus is on the modern era (reference period only goes back to 1971) it might have been better if the authors had used an atmospheric reanalysis that assimilates upper air observations, rather than the 20th century reanalysis, which only assimilates surface pressure. While monthly means are well reproduced in that reanalysis, the results may be less accurate for sub-monthly values. In any event, it might be worth comparing the results in Fig. 7 with e.g. the results based on the older NCAR/NCAR reanalysis just as a sanity check.

**Response:** We apologize for the confusion. The reference to the data sets in our manuscript is misleading. We have actually used the NCEP/NCAR Reanalysis 1 (1948 – 2015). We started our analysis using the 20$^{th}$ century reanalysis, but realizing similar concerns to the ones you mention, all the results and figures in the paper have been obtained based on the NCEP/NCAR Reanalysis 1 (1948 – 2015). We missed this change to the references and have modified the revised manuscript to add the proper set of data.

Other details:
- please check the cost "5000 billion Euros [EEA, 2010]." – line 5, page 2
- line 18, page 3: "management" should be "managed"
- line 22, page 4: "to" should be "the"
- what is the reference period for the SST anomalies in Fig 5a?
- state the reference period for indices in caption of Fig 8 (1950-2000?)
- page 8 lines 25-28, should note that some studies indicate that the role of the Mediterranean Sea was largely passive in 2003 (e.g., Tomassini and Elizalde 2012)
- "Siegfried et al. 2014" reference should be "Schubert et al. 2014".
- Figure S5 "As in Figure 9", but for the period 1950 – 2015. should be "As in Figure 8".

**Response:** All the above suggestions/comments have been accounted for in the revised version of the manuscript.

"Warm Season Subseasonal Variability and Climate Extremes in the Northern Hemisphere: The Role of Stationary RossbyWaves," Schubert, S., H.Wang, and M. Suarez. J. Climate, 24, 4773-4792, 2011

**Anonymous Referee #3**

General:
An in-depth study of the 2015 European drought would be a valuable addition to the literature. This is clearly a strong group of researchers, but I was very underwhelmed by this article. The analysis is almost entirely descriptive and largely consists of maps of precipitation and temperature anomalies, along with accompanying maps of drought indices. All of this information is easily or widely available. Most of the conclusions reached are obvious: the drought was associated with below average precipitation, above average temperature, positive 500Pa height anomalies, and "widespread areas of negative SPI and SPEI" (quote from Abstract). What drought doesn't have these features? The result that is most interesting is the associated sea-surface temperature patterns, but, like the other analyses, this is a largely descriptive exercise. The comparison with 2003 provides some additional substance, but it makes one wonder about the features of other past droughts. Why use just one year for the comparison when an ensemble approach is much more valuable? Similarly, no probabilistic information on how unusual (extreme) the 2015 drought was is provided other than that inferred from the SPI and SPEI values. The article reads like a routine government report rather than cutting-edge research.

**Response:** The aim of this study was to present a broader overview of the drivers of the 2015 summer drought both from a climatological point of view (present paper) as a companion to the hydrological point of view (http://www.hydrol-earth-syst-sci-discuss.net/hess-2016-366/), following the pattern of previous atmosphere-only summaries of individual climatic extreme events. Analyzing and managing drought in a pro-active way requires a concerted action of the hydrological and climatic communities. These twin papers are a first attempt to emphasize the need of such actions from different communities. In this way, the two papers are largely descriptive, but are designed as a clear and comprehensive summary of the 2015 event to promote more detailed future studies by compiling available data and providing a broad analysis to highlight important anomalies and unresolved research issues.
Nevertheless, following the reviewer comments and the goal of providing a "first pass" analysis, we have substantially revised the manuscript, by adding more time series analysis, placing the 2015 European drought into a long term context (see Sections 3.1, 3.2 and 5.2). Moreover, we have extended our analysis regarding the influence of the Mediterranean and North Atlantic Ocean SST on the occurrence of summer droughts, by employing composite maps analysis over the period 1950 - 2015 (section 4.4). This analysis highlights an important topic for future atmosphere-ocean modeling studies to confirm the causal link between Mediterranean SST and summer drought. Future study is also needed to understand whether the North Atlantic SST anomaly, which was a distinctive feature of the 2015 event but shown to have no historical link to European drought, could have a climatological link to drought.
Attached the to our direct response to each of the reviewer comments, is also the revised version of the manuscript will all the corrections/modification we have made highlighted in red.

Specific:
(1) At a minimum, this article needs to incorporate a more quantitative and probabilistic perspective on the 2015 drought. The SPI and SPEI values are a start but don't fully show how unusual the values are. This could be done at each grid point or regionally over appropriate areas (such as agricultural regions or drainage basins).

Response: To provide more quantitative analysis, we have incorporated gridded ranking maps in the revised manuscript, which show where summer 2015 ranks, in terms of amplitude and spatial extent, compared to the last 66 years (see Sections 3.1, 3.2 and 5.2 5.2). We have applied this type of analysis

for almost all the variables involved in our study (e.g. SPI3, SPEI3, SST, P and Tmax). These ranking maps show that summer 2015 stands out as the warmest and driest summer, over the 1950 – 2015 period, for the area stretching from the eastern Czech Republic to Ukraine. Moreover, we show that for Europe, as a whole, summer 2015 it is among the six hottest and driest summers since 1950.

(2) Building on the lack of probabilistic information, there also is an opportunity to include a paleoclimatic drought perspective. This could easily be done by using data from the Old World Drought Atlas (Cook et al., 2015) and then using that information in a more probabilistic approach.

Response: Including a paleo perspective, while interesting and potentially valuable as a follow-up study, is beyond the scope of this paper and somewhat outside the paper's objectives of characterizing the severity, progression, and drivers of the 2015 event. Moreover, the summer of 2015 is not included in the Old World Drought Atlas, which makes bias and spatial inconsistencies between observed and reconstructed scPDSI during 2015 difficult to estimate. Finally, the regions most affected by drought in 2015 (the eastern part of Poland and Ukraine), have poor coverage by tree rings sites in the Old World Atlas scPDSI reconstruction (see Figure 1 in Cook et al. 2015). This suggests that reconstructions for this region would rely on more distant tree-rings, losing some spatial resolution and leading to potential inconsistencies with the observed 2015 data. With additional validation for the region of interest and potentially including tree-rings from 2015, this would be an interesting study, but these additional steps make it outside the scope of an already long study.

(3) In terms of what other droughts have occurred and how they compare to these two, the current analysis suggests that the SST dipole may be an important and presumably causal feature. But it is unclear how often this occurs and how long it persists. Is it necessary but not sufficient? What other SST patterns cause extreme droughts in this area? Some additional analysis of that feature could make this a much more useful piece.

Response: In the new version of the manuscript we have tried to extend the analysis regarding the influence of the Mediterranean and the North Atlantic Ocean SST on the occurrence of summer droughts over the European region. We have extended our analysis over the last 66 years (please see Section 4.1 and the new Section 4.4). By employing composite maps analysis, we show that, over the last 66 years, summer droughts, especially over the eastern part of Europe, occur after and concurrently with a warm western Mediterranean SST and that the cold blob in the North Atlantic basin does not relate significantly with the summer droughts occurrence. We hope this finding based on historical analysis will motivate new sensitivity studies designed to specifically target the conclusions drawn here.

Technical:
(1) In the title, "2015 European drought" seems more appropriate than "European 2015 drought".
(2) The information on losses of 5000 billion Euro is given twice in the Introduction.
(3) The rainbow color map used in Fig. 7 is not appropriate.

Response: All these technical have been accounted for in the revised version of the manuscript.

References:
Cook, Edward R., et al. "Old World megadroughts and pluvials during the Common Era." Science Advances 1.10 (2015): e1500561.

[revised manuscript text omitted]

---

## Author Response (AR2)

Dear Dr. Stahl,

Please find attached the revised version of our manuscript "**The European 2015 drought from a climatological perspective**" by M. Ionita et al.

In this version of the manuscript we have taken into account the requests made by the editor, more specifically:

1. We have change the abstract and made it more compact and easier to follow.

2. We have also tried to use proper names for variables. This is mainly valid for Tmax, which is now Tx.

3. The Discussion and Conclusions part has been separated in two distinct parts: 6. Discussion and 7. Conclusions and recommendations.

In the Conclusions part we highlight the main findings and try to emphasize (recommend) the need for a better communication and joint analysis between the different communities (climate and hydrology).

4. We added a small sub-section with the methods employed in our study (Composite maps and ranking maps).

4. We have reduced the number of figures, in the main document, to 12 and the rest are in the supplementary files.

All authors have agreed with resubmission of this revised manuscript, and no part of the paper is published or under review at another journal.

Sincerely,

Monica Ionita